# circHIPK3 nucleates IGF2BP2 and functions as a competing endogenous RNA

**Trine Line Hauge Okholm[1,2,3]\*, Andreas Bjerregaard Kamstrup[4], Morten Muhlig Nielsen[1,3], Anne Kruse Hollensen[4], Mette Laugesen Graversgaard[4], Matilde Helbo Sørensen[4], Lasse Sommer Kristensen[5], Søren Vang[1], Samuel S Park[6], Eugene Yeo[6], Lars Dyrskjøt[1,3], Jørgen Kjems[4,7], Jakob Skou Pedersen[1,3,8]\*†, Christian Kroun Damgaard[4]\*†**

[1]Department of Molecular Medicine (MOMA), Aarhus University Hospital, Aarhus, Denmark; [2]Departments of Otolaryngology-Head and Neck Surgery and Microbiology & Immunology, University of California, San Francisco, San Francisco, United States; [3]Department of Clinical Medicine, Aarhus University, Aarhus, Denmark; [4]Department of Molecular Biology and Genetics, Aarhus University, Aarhus, Denmark; [5]Department of Biomedicine, Aarhus University, Aarhus, Denmark; [6]Department of Cellular and Molecular Medicine, University of California, San Diego, San Diego, United States; [7]Interdisciplinary Nanoscience Center (iNANO), Aarhus University, Aarhus, Denmark; [8]Bioinformatics Research Center (BiRC), Aarhus University, Aarhus, Denmark

**\*For correspondence:**
trineline.okholm@ucsf.edu (TLHO);
jakob.skou@clin.au.dk (JSP);
ckd@mbg.au.dk (CKD)

†These authors contributed equally to this work

**Abstract** Circular RNAs represent a class of endogenous RNAs that regulate gene expression and influence cell biological decisions with implications for the pathogenesis of several diseases. Here, we disclose a novel gene-regulatory role of circHIPK3 by combining analyses of large genomics datasets and mechanistic cell biological follow-up experiments. Using time-course depletion of circHIPK3 and specific candidate RNA-binding proteins, we identify several perturbed genes by RNA sequencing analyses. Expression-coupled motif analyses identify an 11-mer motif within circHIPK3, which also becomes enriched in genes that are downregulated upon circHIPK3 depletion. By mining eCLIP datasets and combined with RNA immunoprecipitation assays, we demonstrate that the 11-mer motif constitutes a strong binding site for IGF2BP2 in bladder cancer cell lines. Our results suggest that circHIPK3 can sequester IGF2BP2 as a competing endogenous RNA (ceRNA), leading to target mRNA stabilization. As an example of a circHIPK3-regulated gene, we focus on the *STAT3* mRNA as a specific substrate of IGF2BP2 and validate that manipulation of circHIPK3 regulates IGF2BP2-*STAT3* mRNA binding and, thereby, *STAT3* mRNA levels. Surprisingly, absolute copy number quantifications demonstrate that IGF2BP2 outnumbers circHIPK3 by orders of magnitude, which is inconsistent with a simple 1:1 ceRNA hypothesis. Instead, we show that circHIPK3 can nucleate multiple copies of IGF2BP2, potentially via phase separation, to produce IGF2BP2 condensates. Our results support a model where a few cellular circHIPK3 molecules can induce IGF2BP2 condensation, thereby regulating key factors for cell proliferation.

## eLife assessment

This work explores the role of one the most abundant circRNAs, circHIPK3, in bladder cancer cells, showing with **convincing** data that circHIPK3 depletion affects thousands of genes and that those downregulated (including STAT3) share an 11-mer motif with circHIPK3, corresponding to a binding

site for IGF2BP2. The experiments demonstrate that circHIPK3 can compete with the downregulated mRNAs targets for IGF2BP2 binding and that IGF2BP2 depletion antagonizes the effect of circHIPK3 depletion by upregulating the genes containing the 11-mer. These **important** findings contribute to the growing recognition of the complexity of cancer signaling regulation and highlight the intricate interplay between circRNAs and protein-coding genes in tumorigenesis.

## Introduction

Circular RNAs (circRNAs) are endogenous predominantly non-coding RNAs, formed from precursor mRNA through backsplicing events (*Starke et al., 2015*). Although first thought to be by-products or a result of erroneous splicing (*Cocquerelle et al., 1993*), recent improvements in next-generation sequencing of non-polyadenylated transcriptomes and bioinformatics technology have revealed the existence of thousands of human circRNAs (*Jeck et al., 2013*; *Liu and Chen, 2022*; *Memczak et al., 2013*). Now, accumulating evidence suggests that circRNAs regulate numerous cell biological processes (reviewed in *Liu and Chen, 2022*) and have been implicated in tumorigenesis (*Chen et al., 2019a*; *Vo et al., 2019*). Some circRNAs possess biomarker potential (reviewed in *Kristensen et al., 2022*; *Zhang et al., 2018*) owing to expression specificity (*Memczak et al., 2013*), structural stability (*Jeck et al., 2013*), and abundance in exosomes (*Li et al., 2015b*), blood (*Memczak et al., 2015*), plasma (*Bahn et al., 2015*), and urine (*Vo et al., 2019*).

The functional impact of circRNAs in cancer is still a major area of research. Many circRNAs can interact with RNA-binding proteins (RBPs) (*Ashwal-Fluss et al., 2014*; *Du et al., 2016*) and several circRNAs are suggested to function as miRNA sponges (*Li et al., 2015a*; *Li et al., 2017*; *Zheng et al., 2016*). However, besides ciRS-7 and circSRY, which can bind a large number of miRNAs (*Hansen et al., 2013*; *Memczak et al., 2013*), circRNAs are not enriched for more miRNA-binding sites than would be expected by chance (*Guo et al., 2014*). Recently, it was discovered that some circRNAs can be translated into proteins under certain conditions (*Chen et al., 2021*; *Petrucci et al., 2017*; *Pamudurti et al., 2017*) but the majority of circRNAs likely do not encode proteins (*Stagsted et al., 2019*). Only a few circRNAs have been intensely studied and functionally described in vivo, for example, ciRS-7 (*Piwecka et al., 2017*). However, functional implications are usually restricted to specific conditions, cells, and tissues, with small datasets and few experimental validation studies being general caveats that provide inconsistent conclusions. Additionally, there is a general lack of knowledge on functional mechanisms, beyond potential miRNA sponging, which is often based on studies of overexpression of circRNAs and/or miRNA mimics and reporter assays.

CircHIPK3 (chr11:33307958–33309057) (hg19) is the most prevalent alpha circRNA, that is, it is the circRNA most often found to be highest expressed in human samples (*Stagsted et al., 2019*). It originates from the second exon of the gene *HIPK3* and possesses highly conserved regions across species (*Zheng et al., 2016*). Accumulating evidence indicates that circHIPK3 possesses biomarker potential and could be a potential target in cancer therapy (reviewed in *Wen et al., 2020*; *Xie et al., 2020*). While circHIPK3 is often deregulated in cancer, studies indicate that circHIPK3 play dual roles in tumorigenesis; some studies report that circHIPK3 is downregulated in cancers and possess tumor suppressor functions, for example, in bladder cancer (BC) (*Li et al., 2017*; *Okholm et al., 2017*), ovarian cancer (*Teng et al., 2019*), and gastric cancer (*Ghasemi et al., 2019*), while other studies describe the opposite effect of circHIPK3 in the same and other cancer types, for example, glioma cancer (*Jin et al., 2018*), colorectal cancer (*Zeng et al., 2018*), prostate cancer (*Cai et al., 2019*), ovarian cancer (*Liu et al., 2018*), lung cancer (*Chen et al., 2020*), gastric cancer (*Cheng et al., 2018*), and hepatocellular carcinoma (*Chen et al., 2018*; *Chen et al., 2020*; *Zheng et al., 2016*). Several of these studies conclude that circHIPK3 regulates a wide range of genes by sponging different miRNAs in various cancer tissues, for example, miR-124-3p (*Chen et al., 2018*; *Chen et al., 2020*; *Zheng et al., 2016*), miR-338-3p (*Cai et al., 2019*), miR-193a-3p (*Chen et al., 2019b*), miR-7 (*Zeng et al., 2018*), miR-654 (*Jin et al., 2018*), and miR-558 (*Li et al., 2017*).

In a previous study, we found that circHIPK3 is expressed in patients with non-muscle invasive bladder cancer (NMIBC) and correlates positively with BC progression independently of the parent gene (*Okholm et al., 2017*). Additionally, we found that circHIPK3 is higher expressed than the corresponding linear transcript and possesses key biological features, for example, enrichment of evolutionary conserved regions and miRNA target sites, indicating a regulatory function of its own.

Here, we extensively evaluate the functional impact of circHIPK3 in BC cells by combining large-scale computational analyses with comprehensive validation experiments. By conducting time-series knockdown (KD) experiments, we identify thousands of genes that are deregulated upon circHIPK3 KD across different BC cell lines. We find no evidence for a miRNA-sponge function of circHIPK3, since predicted miRNA-targeted mRNAs are not significantly affected by efficient circHIPK3 depletion. Instead, using a computational method, Regmex, that evaluates the enrichment of nucleotide motifs in deregulated genes (*Nielsen et al., 2018*), we identify an 11-mer circHIPK3 motif that is also enriched in mRNAs that become downregulated upon circHIPK3 KD. We examine a large eCLIP dataset of RBP targets and discover that the motif is part of a binding site for IGF2BP2. By performing RNA immunoprecipitation (RIP), we demonstrate that IGF2BP2 associates with circHIPK3, primarily via the 11-mer motif. We show that IGF2BP2 KD leads to upregulation of genes containing the 11-mer motif and that IGF2BP2 KD counteracts the effect of circHIPK3 KD on target genes and proliferation pathways. One of these targets, *STAT3*, is downregulated upon circHIPK3 KD, while TP53, a known target of STAT3 inhibition, becomes upregulated. Consistent with a circHIPK3 competing endogenous RNA (ceRNA) model, we confirm that *STAT3* mRNA is bound more efficiently by IGF2BP2 upon circHIPK3 depletion. Conversely, this competition is dependent on the 11-mer motif since its presence within circHIPK3 reduces the *STAT3* mRNA–IGF2BP2 interaction. However, careful quantifications of copy numbers of circHIPK3 and IGF2BP2 suggest that a simple ceRNA model is unlikely as IGF2BP2 outnumbers circHIPK3 by orders of magnitude. Since ectopic expression of circHIPK3 induces IGF2BP2 condensation within cells, we propose that even a few circHIPK3 molecules can function as a ceRNAs by nucleating numerous IGF2BP2 molecules per circRNA. Finally, by analyzing large patient cohorts with rich clinical annotations, we show that the expression of circHIPK3 is positively associated with overall survival in BC patients.

## Results

### Thousands of genes are deregulated upon circHIPK3 KD

To comprehensively study the role of circHIPK3 in the regulation of gene expression in BC, we performed time-series KD experiments in BC cell lines followed by RNA sequencing (RNA-Seq). First, we evaluated the expression of circHIPK3 (the second exon of *HIPK3*, position chr11:33,307,958–33,309,057 (hg19), *Figure 1A*) and the corresponding linear transcript in non-malignant (*n* = 4) and metastatic (*n* = 7) BC cell lines (*Hedegaard et al., 2016*; *Okholm et al., 2017*) using the CIRI2 pipeline (*Gao et al., 2018*). In all cell lines except for one, circHIPK3 is higher expressed than the linear transcript (*Figure 1B*). Based on circHIPK3 expression and cell line stability, we chose J82 and UMUC3 for circHIPK3 KD experiments. We designed an siRNA against the backsplicing junction of circHIPK3 to specifically target the circle but not the corresponding linear transcript (*Figure 1A*, Methods). We transfected cells with the circHIPK3-specific siRNA or a scrambled (scr) siRNA as negative control and measured the KD efficiency of circHIPK3 using divergent primers against the unique backsplicing junction (*Figure 1C* and *Supplementary file 1*). As expected, the circHIPK3 siRNA did not lead to co-knockdown of the linear HIPK3 mRNA or HIPK3 protein (*Figure 1—figure supplement 1A, B*). Next, we harvested RNA at five time points (0, 12, 24, 48, and 72 hr post siRNA transfection) in three biological replicates and quantified mRNA expression using QuantSeq (*Moll et al., 2014*). Additionally, we included untreated cells to ensure that gene perturbation is not a consequence of off-target effects or transfection per se (*Figure 1—figure supplement 1*). Based on mRNA expression, we assessed overall sample similarity. As expected, samples cluster according to time (*Figure 1D*).

Next, we performed differential expression analyses and identified thousands of genes that are differentially expressed (DE) upon circHIPK3 KD across time in UMUC3 (*n* = 3072) and J82 (*n* = 2389) (false discovery rate [FDR] <0.1, Benjamini–Hochberg correction), with a large overlap (*n* = 1104) between cell lines (p < 2.2e−16, Fisher's exact test, *Figure 1E*). While few genes are DE at 12 hr, the number of DE genes increases dramatically at 24 hr (*Figure 1F* and *Figure 1—figure supplement 1*), where 1241 (UMUC3) and 642 (J82) genes are significantly down- or upregulated (Wald test, Benjamini–Hochberg correction with FDR <0.1, *Figure 1G*). Convincingly, 92% of the shared DE genes between UMUC3 and J82 cells show the same perturbation profile in both cell lines (*Figure 1—figure supplement 1*), providing reliability of DE genes and indicating shared functions of circHIPK3 across BC cell lines.

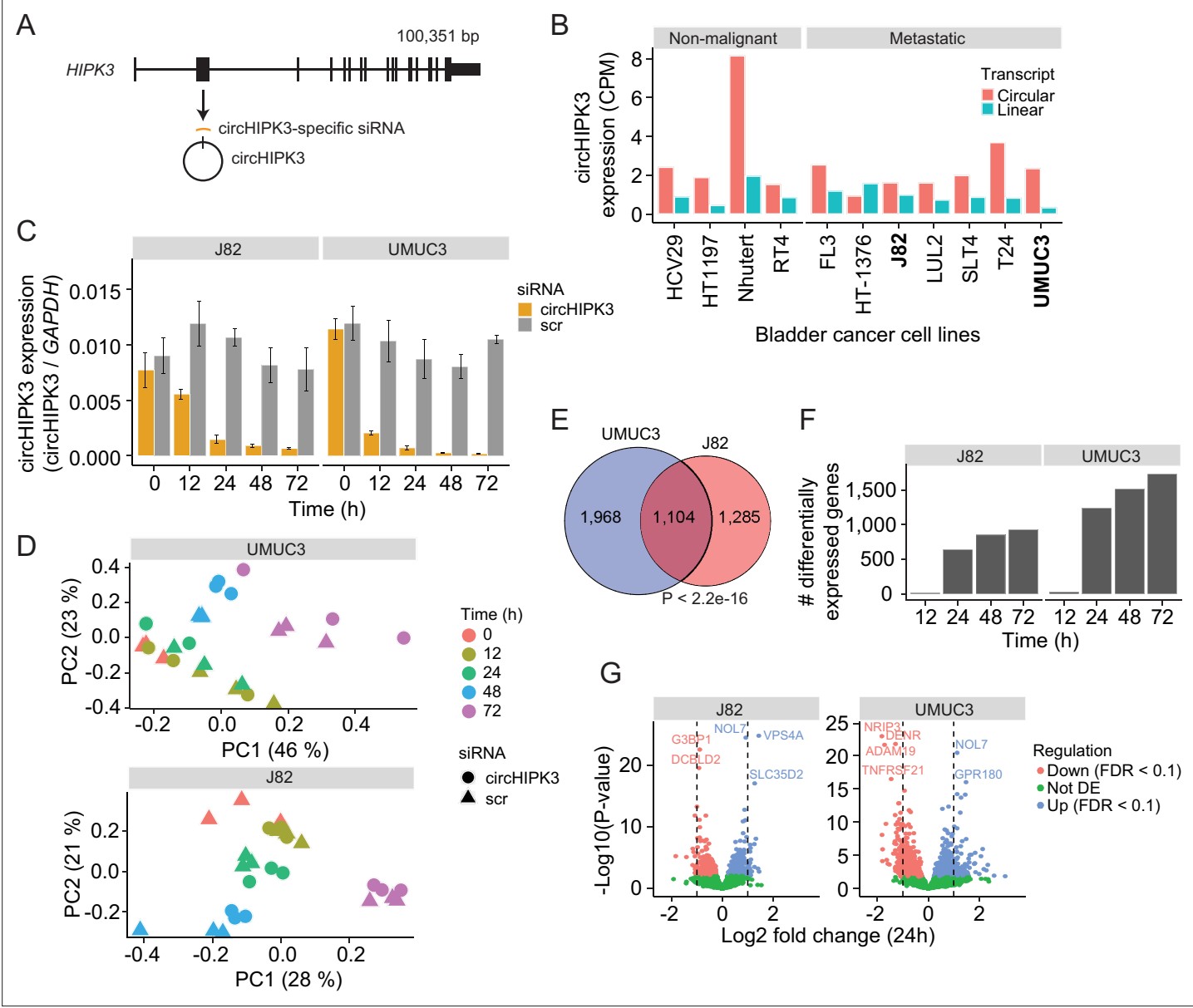

**Figure 1.** Thousands of genes are deregulated upon circHIPK3 knockdown (KD). (**A**) circHIPK3 is produced from the second exon of the *HIPK3* gene. We designed a circHIPK3-specific siRNA targeting the backsplice junction of circHIPK3 to specifically KD the expression of circHIPK3. (**B**) Expression of circHIPK3 and the corresponding linear transcript in 11 bladder cancer cell lines. UMUC3 and J82 were chosen for KD experiments based on circHIPK3 expression levels and cell line stability. CPM = counts per million. (**C**) KD efficiency of circHIPK3 in UMUC3 and J82 at different time points post circHIPK3 (yellow) or scramble (gray) siRNA transfection. Expression is normalized to glyceraldehyde-3-phosphate dehydrogenase (GAPDH) levels. (**D**) Principal component analysis (PCA) plot of gene expression in UMUC3 and J82. PCA plots are based on the genes with the 50% most variance across all cell line samples. Gene expression is log transformed (natural logarithm) and added a pseudocount of 1. (**E**) Overlap of differentially expressed genes across all time points and/or conditions between UMUC3 (*n* = 3072) and J82 (*n* = 2389). p-value obtained by Fisher's exact test. (**F**) Number of differentially expressed genes at each time point (Scr vs circHIPK3) (Wald test, Benjamini–Hochberg correction with false discovery rate [FDR] <0.1). Only genes with perturbed expression profiles across time and/or conditions in UMUC3 (*n* = 3072) or J82 (*n* = 2389) are considered. (**G**) Differential expression analysis between circHIPK3 KD and scr siRNA samples 24 hr post-transfection (Wald test). The log2 fold changes (circHIPK3 KD vs scr) are plotted against the negative log10(p-values). Colors indicate if genes are significantly down- (red) or upregulated (blue) or not differentially expressed (Not DE, green) after Benjamini–Hochberg correction, FDR <0.1. Vertical lines indicate a log2FC >1 or <−1.

The online version of this article includes the following figure supplement(s) for figure 1:

**Figure supplement 1.** Thousands of genes are deregulated upon circHIPK3 knockdown (KD).

## Lack of evidence for miRNA-sponge functions

The most widespread proposed function of circRNAs is as miRNA sponges although this is in many cases based on circRNA/miRNA mimic overexpression studies or reporter assays, which may not provide concluding evidence for an endogenous function (*Jarlstad Olesen and S Kristensen, 2021*). Nevertheless, several studies have suggested that circHIPK3 sponges multiple miRNAs in various tissues (*Chen et al., 2018*; *Jin et al., 2018*; *Li et al., 2017*; *Shan et al., 2017*; *Stoll et al., 2018*; *Zeng et al., 2018*; *Zheng et al., 2016*). Based on AGO-CLIP data of experimentally defined miRNA-binding sites, we previously identified 11 distinctly conserved miRNA target sites in circHIPK3 (*Hamilton et al., 2016*; *Hamilton et al., 2013*; *Okholm et al., 2017*; *Figure 2A* and *Supplementary file 2*) constituting binding sites for 23 miRNAs (circHIPK3-miRNAs). We profiled the expression of ~790 miRNAs in J82 (*Figure 2—figure supplement 1A* and *Supplementary file 3*) and evaluated if the 3′UTRs of downregulated genes upon circHIPK3 KD are enriched with target sites for expressed circHIPK3-miRNAs in J82. Neither the target sites for the highest expressed miRNA (miR-148-3p; CAGUGCA) nor any of the other circHIPK3-miRNAs are enriched among downregulated genes ($p > 0.05$, Chi-square test, *Figure 2B* and *Figure 2—figure supplement 1B, C*), indicating that circHIPK3 does not sponge these miRNAs.

Abundance and regulatory roles of miRNAs are highly cell type and tissue dependent (*Landgraf et al., 2007*; *Ludwig et al., 2016*). Therefore, circHIPK3 could bind different miRNAs in BC cell lines than the ones proposed by the AGO-CLIP data. To evaluate this, we enumerated all possible unique 7-mers ($n = 1032$) in the sequence of circHIPK3 (circHIPK3 7-mers). Of these, 148 correspond to target sites for miRNAs (miRNA 7-mers). No circHIPK3 7-mers occur more than three times.

If circHIPK3 functions as a miRNA sponge, we predict that target genes of the miRNA become downregulated upon circHIPK3 KD. We used a computational method, Regmex (*Nielsen et al., 2018*), to evaluate the enrichment of circHIPK3 7-mers in the 3′UTRs of deregulated genes upon circHIPK3 KD (*Figure 2C* and *Supplementary file 4*). Regmex assigns a positive score to motifs enriched in 3′UTRs of upregulated genes and, correspondingly, a negative score to motifs enriched in 3′UTRs of downregulated genes. In line with perturbed gene expression profiles, the most extreme motif scores were seen 24 hr post-transfection in UMUC3, while motif perturbation in J82 was seen across multiple time points (*Figure 2—figure supplement 1D*). Although motif scores were slightly lower for miRNA 7-mers than for 7-mers that do not constitute a miRNA target site, the median motif score was above 0 and the effect size small in both cell lines (effect size $<0.25$, $p < 0.05$, Wilcoxon Rank Sum Test, *Figure 2—figure supplement 1E*), indicating that 7-mers constituting miRNA target sites are not overall enriched in downregulated genes. Specifically, the miRNA-7-mer with the most negative motif score in both cell lines constitute a target site for miR-1207-5p, however, miR-1207-5p is not conserved and has no reported functional roles. Taken together, we found no clear evidence for circHIPK3 to possess miRNA-sponge functions in our experiments, although this has previously been suggested for circHIPK3 in numerous studies (*Cai et al., 2019*; *Chen et al., 2019b*; *Chen et al., 2018*; *Chen et al., 2020*; *Jin et al., 2018*; *Zeng et al., 2018*; *Zheng et al., 2016*; *Zhou et al., 2021*; *Cai et al., 2019*; *Chen et al., 2019b*; *Chen et al., 2018*; *Chen et al., 2020*; *Jin et al., 2018*; *Zeng et al., 2018*; *Zheng et al., 2016*).

## A long motif in circHIPK3 is enriched in downregulated genes upon circHIPK3 KD

Some circRNAs have been shown to interact with other regulatory factors like RBPs (*Ashwal-Fluss et al., 2014*; *Du et al., 2016*; *Hollensen et al., 2020*). While miRNAs typically rely on perfect pairing of the miRNA seed region to the target site (*Bartel, 2009*), RBP-binding sites are generally less well-defined as RBP binding often depends on contextual properties, including secondary structure and nucleotide compositions (*Dominguez et al., 2018*).

We assessed the ability of circHIPK3 to regulate target genes through other regulatory factors than miRNAs by examining the expression correlation of all circHIPK3 7-mers (*Figure 2D*). Interestingly, the sequence composition of nine of the top ten 7-mers with the most negative motif scores in UMUC3 are shifted by one base or only differ by a single or two bases. These 7-mers are also among the most negatively scoring motifs in J82 cells (*Figure 2—figure supplement 1F*), suggesting downregulation of motif-containing mRNAs upon circHIPK3 KD. Aligning these nine 7-mers to circHIPK3 reveals a larger 11-mer within circHIPK3, AGGCCCCC(A/T)GC (*Figure 2E*). Allowing up to two base

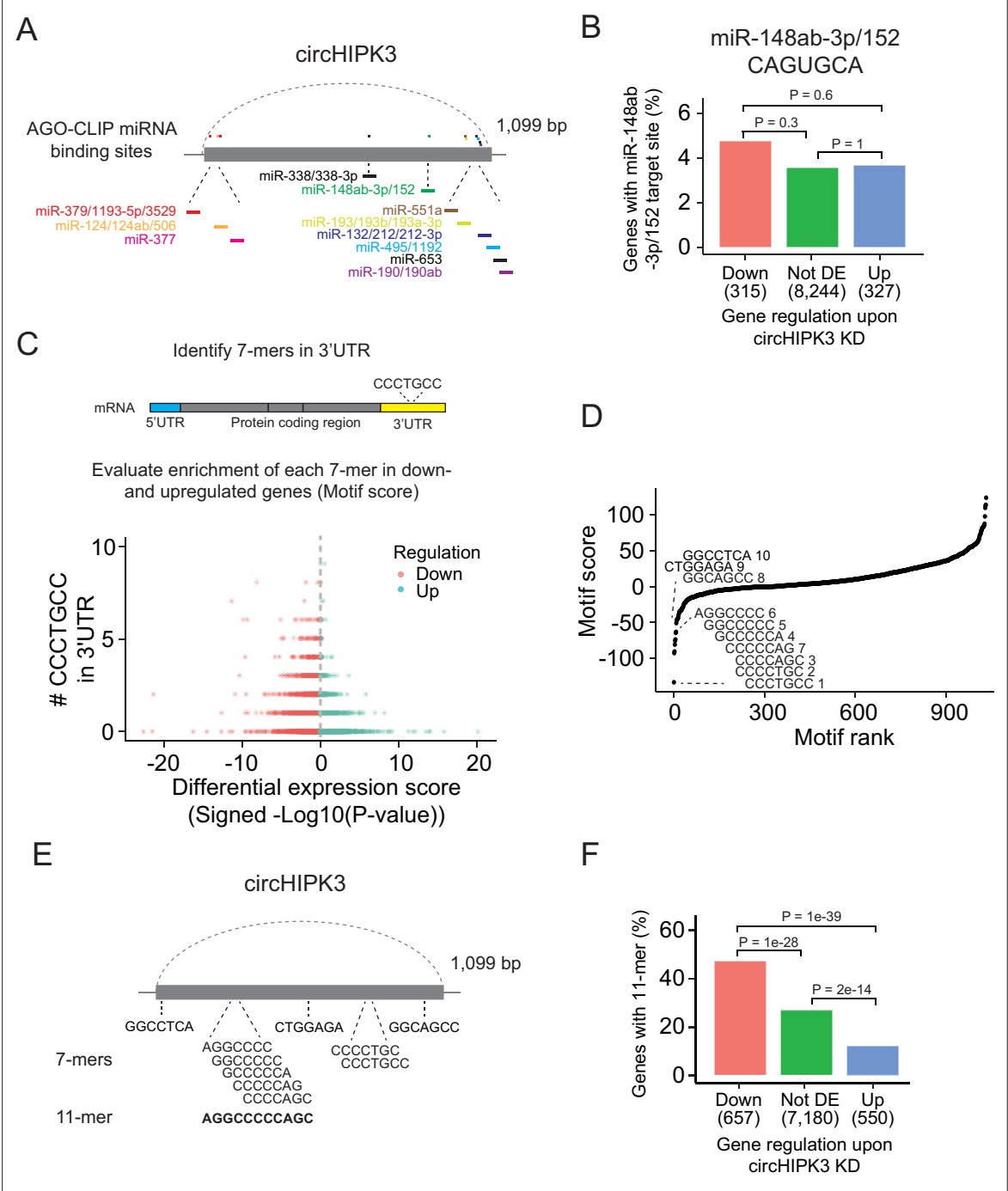

**Figure 2.** A long motif in circHIPK3 is enriched in downregulated genes upon circHIPK3 knockdown (KD). (**A**) Illustration of conserved miRNA-binding sites in circHIPK3 based on AGO-CLIP data. (**B**) Percentage of genes in each group with miR-148ab-3p/152 target sites (CAGUGCA) in their 3'UTRs. Gene regulation is based on circHIPK3 KD vs scr siRNA transfection 24 hr post-transfection in J82. p-values obtained by Chi-square test. (**C**) Procedure for motif enrichment analysis using Regmex. First, we extract all 7-mers in the sequence of circHIPK3. For each 7-mer, for example, CCCTGCC, we identify their presence in the 3'UTR of genes. We order all genes according to a differential expression score calculated as the −log10(p-value) multiplied by the fold change direction, for example, 1 for upregulated genes and −1 for downregulated genes. Then we calculate a motif score for each 7-mer based on their occurrences in either down- or upregulated genes. If a 7-mer has a positive motif score it means that it is enriched in the 3'UTR of genes that are upregulated upon circHIPK3 KD. Conversely, 7-mers with a negative motif score are primarily found in genes that are downregulated upon circHIPK3 KD. The 7-mer, CCCTGCC, is used for illustration purposes. (**D**) Regmex motif scores for circHIPK3 7-mers (UMUC3, 24 hr). Alignment of the ten 7-mers with the most negative motif scores are shown. 7-mers are ranked from most negative to most positive motif scores. Numbers correspond to rank. (**E**) Illustration of circHIPK3 and position of the ten motifs with the lowest motif scores. The 7-mers with the most negative

*Figure 2 continued on next page*

Figure 2 continued

motif scores found in circHIPK3 comprise a larger 11-mer, AGGCCCCCAGC, present in the sequence of circHIPK3. (**F**) Percentage of genes in each group containing the 11-mer motif upon circHIPK3 KD in UMUC3 cells (24 hr). p-values obtained by Chi-square test.

The online version of this article includes the following figure supplement(s) for figure 2:

**Figure supplement 1.** A long motif in circHIPK3 is enriched in downregulated genes upon circHIPK3 knockdown (KD).

substitutions, we evaluated the presence of the 11-mer motif in the 3'UTR of genes. Interestingly, the motif is enriched in downregulated genes compared to upregulated genes (p < 1e−39) and genes that are not differentially expressed (not DE) (p < 1e−28) in both UMUC3 (Chi-square test, *Figure 2F*) and J82 cells (p < 0.0001 for both comparisons, Chi-square test, *Figure 2—figure supplement 1G*). This finding was further validated in a third BC cell line FL3 (p < 2e−39 for both comparisons, Chi-square test, *Figure 2—figure supplement 1H*).

Our results show that the 7-mers with the most negative motif scores comprise a larger circHIPK3 motif with some variability and suggest that circHIPK3 regulate target genes containing this motif. Thus, we hypothesized that the 11-mer motif in circHIPK3 comprise a binding site for RBPs that may be crucial for its function, potentially by sponging or otherwise regulating specific RBPs.

## The 11-mer motif in circHIPK3 constitutes a binding site for IGF2BP2

Next, we evaluated if the 11-mer motif represents a binding site for RBPs. Since RBP targets are generally poorly defined and several factors influence RBP binding, for example, contextual features and expression of cofactors, sequence-based analyses are likely not sufficient to capture true RNA–RBP interactions. Therefore, to identify RBPs that target the 11-mer in circHIPK3, we analyzed a comprehensive eCLIP dataset with experimentally defined binding sites of 150 RBPs in HepG2 and K562 cells (*Van Nostrand et al., 2016*). This revealed high confidence binding sites of several RBPs in the circHIPK3 loci (circHIPK3–RBPs) (*Figure 3A*). Specifically, binding sites for GRWD1 (HepG2) and IGF2BP2 (K562) overlap the 11-mer.

To interact, both circHIPK3 and RBPs should be expressed in the same subcellular compartments of the cells. We analyzed circHIPK3 localization in total RNA-Seq datasets of fractionated cells (*The ENCODE Project Consortium, 2012*) and found that circHIPK3 is highly expressed in the cytoplasm of HepG2 and K562 cells (*Figure 3—figure supplement 1A*), which corroborates our previous findings in the BC cell lines T24, HCV29, and FL3 (*Okholm et al., 2017*). From immunofluorescence imaging of RBP localization in HepG2 (*Van Nostrand et al., 2020*), we found that all circHIPK3–RBPs occupy the cytoplasmic fraction of the cell, except for U2AF1 and U2AF2, which span the intron–exon boundary at 3' splice sites and are instrumental for mRNA splicing (*Supplementary file 5*). To confirm circHIPK3–RBP interactions, we performed RIP of identified circHIPK3–RBPs as well as an unrelated control RBP, UCHL5. Using divergent primers targeting the unique backsplice junction of circHIPK3 (*Supplementary file 1*), we specifically verified circHIPK3 pull-down in IGF2BP1 and IGF2BP2 RIP experiments by semi-quantitative reverse transcription polymerase chain reaction (RT-PCR) in both cell lines (*Figure 3B*). Faint bands indicate that circHIPK3 also binds other RBPs albeit with lower affinity. Notably, circHIPK3 interacts weakly with GRWD1 in HepG2 but not in K562, as suggested by the eCLIP data.

Studies have shown that circRNAs that serve as miRNA sponges compete for binding of miRNAs and thereby influence abundance of the corresponding target mRNAs (*Hansen et al., 2013*). If circHIPK3 competes for RBP binding in a similar manner, we would expect deregulated genes upon circHIPK3 KD to contain RBP-binding sites as well. Based on the eCLIP data, we identified all transcriptome-wide binding sites of the RBPs predicted to interact with circHIPK3 (*Figure 3A*). Focusing on K562-RBPs, we found that while genes that become upregulated upon circHIPK3 KD generally contain more circHIPK3–RBP-binding sites (*Figure 3—figure supplement 1B*), IGF2BP2-binding sites are enriched in genes that become downregulated upon circHIPK3 KD in both UMUC3, J82, and FL3 (all p < 0.05, Chi-square tests, *Figure 3C* and *Figure 3—figure supplement 1C*). This suggests that IGF2BP2 may become sequestered by circHIPK3 via the 11-mer motif and allow for stabilization of target mRNAs.

We evaluated how often binding sites of circHIPK3–RBPs overlap the 11-mer motif and found that this is more often the case for IGF2BP2-binding sites than binding sites of the other circHIPK3–RBPs when scrutinizing K562 datasets (*Figure 3—figure supplement 1D*). Furthermore, IGF2BP2-11-mer

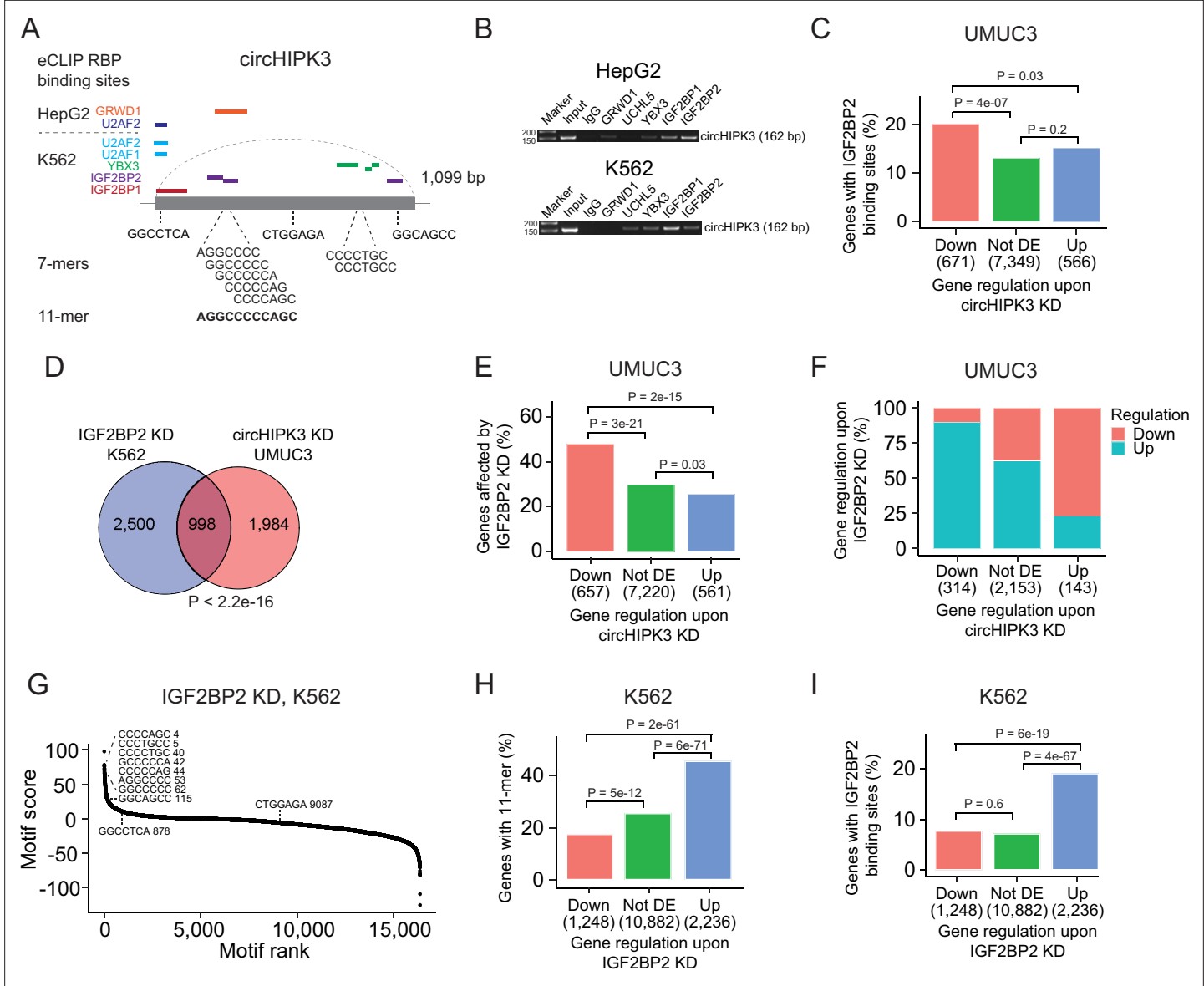

**Figure 3.** The 11-mer motif in circHIPK3 constitutes a binding site for IGF2BP2. (**A**) Illustration of RNA-binding protein (RBP)-binding sites in circHIPK3 based on eCLIP data in the ENCODE cell lines HepG2 and K562. (**B**) RNA immunoprecipitation of circHIPK3–RBPs and others in HepG2 and K562 cells. Bands semi-quantitatively confirm that circHIPK3 interacts with IGF2BP1 and IGF2BP2. Marker indicates 50 bp. (**C**) Percentage of genes in each group containing IGF2BP2-binding sites (K562) in UMUC3 (24 hr). p-values obtained by Chi-square test. (**D**) Overlap between genes that are affected by circHIPK3 knockdown (KD) in UMUC3 and IGF2BP2 KD in K562. p-value obtained by Fisher's exact test. (**E**) Percentage of genes in each group affected by IGF2BP2 KD in K562. p-values obtained by Chi-square test. (**F**) Regulation of genes affected by both circHIPK3 KD (UMUC3, 24 hr) and IGF2BP2 KD (K562). The x-axis indicates gene regulation upon circHIPK3 KD in UMUC3. Percentage on y-axis and colors show how these genes are regulated upon IGF2BP2 KD. Downregulated genes upon circHIPK3 KD are mainly upregulated upon IGF2BP2 KD and vice versa. (**G**) Regmex motif enrichment analysis upon IGF2BP2 KD in K562. 7-mers are ranked from most positive to most negative motif scores. The ten 7-mers with the most negative motif scores in UMUC3 are shown. Numbers indicate motif rank. All possible 7-mers are evaluated (n = 16,384). Percentage of genes in each group containing the 11-mer motif (**H**) and IGF2BP2-binding sites (**I**) upon IGF2BP2 KD in K562. p-values obtained by Chi-square test.

The online version of this article includes the following source data and figure supplement(s) for figure 3:

**Source data 1.** Original file of the agarose gel shown in *Figure 3B*.

**Figure supplement 1.** The 11-mer motif in circHIPK3 constitutes a binding site for IGF2BP2.

overlap is enriched in downregulated genes (p < 0.01, Chi-square test, *Figure 3—figure supplement 1E*). We found no enrichment of HepG2-RBPs in downregulated genes (*Figure 3—figure supplement 1F*).

## IGF2BP2 KD counteracts the effect of circHIPK3 KD

If circHIPK3 functions as a ceRNA for IGF2BP2, downregulation of IGF2BP2 would be expected to have the opposite effect of circHIPK3 KD on target genes. Therefore, we evaluated if genes affected by circHIPK3 KD in BC cell lines are also affected in ENCODE experiments of IGF2BP2 KD and found a significant overlap of altered genes (p < 2.2e−16, Fisher's exact tests, *Figure 3D* and *Figure 3—figure supplement 1G*). The overlap was largest for downregulated genes in both UMUC3 (p < 2e−15, Chi-square test, *Figure 3E*) and J82 cells (p < 0.001, Chi-square test, *Figure 3—figure supplement 1H*). Interestingly, the genes that are downregulated by circHIPK3 KD are primarily upregulated upon IGF2BP2 KD suggesting an mRNA destabilization function of IGF2BP2 (*Figure 3F* and *Figure 3—figure supplement 1I*). RegMex analysis of all possible 7-mers showed that the 7-mers that constitute the 11-mer motif in circHIPK3 are among the highest scoring motifs upon IGF2BP2 KD in K562 (*Figure 3G*). Accordingly, the 11-mer motif and IGF2BP2-binding sites are highly enriched in upregulated genes (p < 6e−19, Chi-square test, *Figure 3H, I*). While there were no IGF2BP2 eCLIP data available in HepG2, analogous enrichments were observed in upregulated genes upon IGF2BP2 KD in HepG2, suggesting a similar mechanism in HepG2 (*Figure 3—figure supplement 1J,K*).

Taken together, our results suggest that the 11-mer motif in circHIPK3 constitutes a binding site for IGF2BP2 and that IGF2BP2-binding sites and the 11-mer motif are enriched in downregulated mRNA transcripts upon circHIPK3 KD, while the opposite is observed upon IGF2BP2 KD. Furthermore, we find that IGF2BP2 KD counteracts the effect of circHIPK3 KD on target mRNAs, indicating that the circHIPK3–IGF2BP2 complex positively regulates mRNAs targeted by IGF2BP2.

## circHIPK3 interacts with IGF2BP2 and affects genes controlling cell cycle progression

To verify circHIPK3–IGF2BP2 interactions in BC we performed RIP experiments in the BC cell line FL3 for the two RBPs that have binding sites across the 11-mer: IGF2BP2 and GRWD1. Additionally, we included IGF2BP1, which has binding sites across one of the downregulated 7-mers constituting the 11-mer motif and several binding sites in the 3'UTR of the parent gene. Consistent with our findings above, circHIPK3 interacts strongly with IGF2BP2 in BC cells but not with GRWD1 (*Figure 4A* and *Figure 4—figure supplement 1A*). Importantly, IGF2BP2 interacts with *ACTB* mRNA with much lower affinity (*Figure 4A* and *Figure 4—figure supplement 1A*). CircHIPK3 also interacts with IGF2BP1, albeit with lower binding affinity than observed for IGF2BP2. RIP experiments of IGF2BP2 upon circHIPK3 overexpression, using Laccase2-driven circRNA expression (*Kramer et al., 2015*), confirmed tight and specific binding between circHIPK3 and IGF2BP2 (p < 0.002, *T*-test, *Figure 4B*).

To confirm regulatory implications of circHIPK3–IGF2BP2 binding in BC, we evaluated the presence of the 11-mer motif in deregulated genes in previously performed experiments of IGF2BP2 KD in BC cells (*Okholm et al., 2020*). Consistent with our observations in K562 and HepG2, we found that the 11-mer motif is enriched in upregulated genes compared to downregulated genes upon IGF2BP2 KD in UMUC3 (p < 7e−04, Chi-square test, *Figure 4C*) and J82 cells (p < 0.002, Chi-square test, *Figure 4—figure supplement 1B*).

IGF2BP2 is a post-transcriptional regulator involved in many mRNA processes, including mRNA stability, transportation, localization, and translation (reviewed in *Cao et al., 2018*). Using a peptide spanning the third and fourth K-homology domain of IGF2BP2 (KH34 domains), a recent SELEX study identified a bipartite recognition element (RE), which confers high affinity RNA binding: CUCAC-(N10-15)-(A/U)-GG-(A/U) (*Biswas et al., 2019*). Here, we found a significant overlap between genes containing the 11-mer motif and the IGF2BP2 KH34 RE in their 3'UTRs (p < 2.2e−16, Fisher's exact test, *Figure 4—figure supplement 1C*). Additionally, the IGF2BP2 KH34 RE is enriched in downregulated genes upon circHIPK3 KD in UMUC3 cells (p < 0.002, Chi-square test, *Figure 4—figure supplement 1D*).

Studies have shown that IGF2BP2 is overexpressed in many cancers, correlates with progression, and promotes cancer cell proliferation and migration (reviewed in *Cao et al., 2018*), while we and

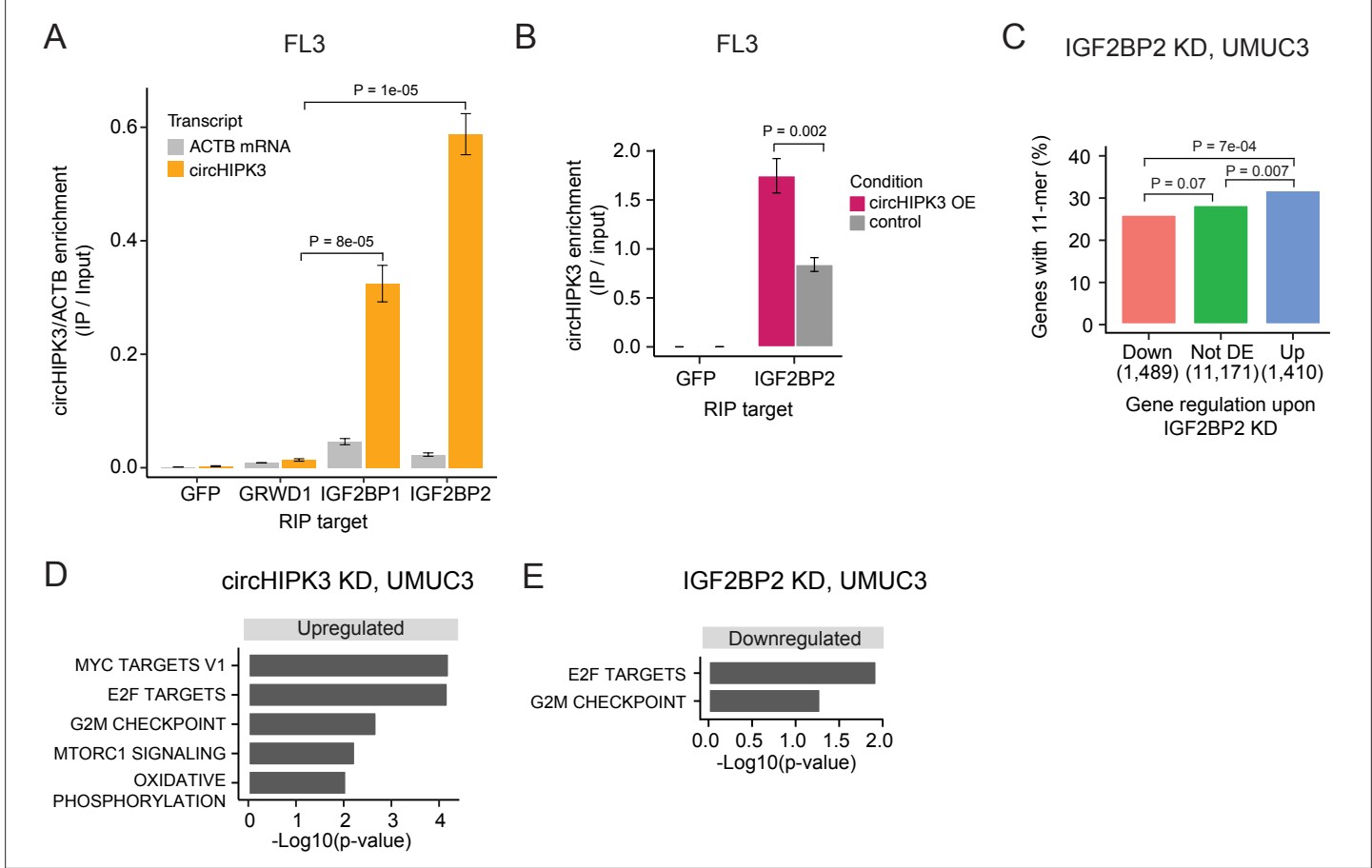

**Figure 4.** circHIPK3 interacts with IGF2BP2 and affects genes controlling cell cycle progression. (**A**) Relative enrichment of circHIPK3 levels or *ACTB* mRNA between IP and input for the three RNA-binding proteins (RBPs) GRWD1, IGF2BP1, and IGF2BP2 in the bladder cancer cell line FL3. Green fluorescent protein (GFP) was used as a negative control. The statistical difference in circHIPK3 enrichment for GRWD1 and IGF2BP1 or IGF2BP2 is indicated by the p-value (*T*-test). (**B**) Relative enrichment of circHIPK3 levels between IP and input for IGF2BP2 upon circHIPK3 overexpression (OE – 24 hr). GFP was used as a negative control. Error bars reflect standard deviation of biological triplicates. (**C**) Percentage of genes in each group containing the 11-mer motif upon IGF2BP2 knockdown (KD) in UMUC3. p-values obtained by Chi-square test. (**D**) Gene set enrichment analysis of 50 hallmarks of cancer upon circHIPK3 KD (24 hr) in UMUC3 cells (false discovery rate [FDR] <0.1 for all shown pathways). (**E**) Downregulated hallmarks of cancer upon IGF2BP2 KD in UMUC3 (p-value <0.05 for all shown pathways).

The online version of this article includes the following source data and figure supplement(s) for figure 4:

**Figure supplement 1.** circHIPK3 interacts with IGF2BP2 and affects genes controlling cell cycle progression.

**Figure supplement 1—source data 1.** Original file for the western blot analysis shown in *Figure 4—figure supplement 1* (anti-HuR).

**Figure supplement 1—source data 2.** Original file for the western blot analysis shown in *Figure 4—figure supplement 1* (anti-streptag).

others have reported that circHIPK3 is downregulated in BC (*Okholm et al., 2017*) and possess tumor suppressor functions (*Teng et al., 2019*).

To assess the functional impact of circHIPK3 and IGF2BP2 in bladder carcinogenesis, we performed gene set enrichment analysis of 50 hallmarks of cancer gene sets (*Liberzon et al., 2015*) and found that several proliferation pathways, for example, E2F targets and G2/M checkpoint, are upregulated upon circHIPK3 KD in both UMUC3 (p < 0.05, *T*-test, *Figure 4D*) and J82 cells (p < 0.05, *T*-test, *Figure 4—figure supplement 1E*). In contrast, these pathways are downregulated upon IGF2BP2 KD (p < 0.05, *T*-test, *Figure 4E*, *Figure 4—figure supplement 1F*), suggesting that the circHIPK3–IGF2BP2-target axis may affect cell cycle progression. However, using an 5'-ethynyl-2'deoxyuridine (EdU)-flowcytometry assay (*Figure 4—figure supplement 1G*), we found only subtle differences in the distribution between G1-, S-, or G2/M-phases of the cell cycle, when measuring these parameters upon KD of either circHIPK3 or IGF2BP2 in FL3 cells.

We observed no difference in *IGF2BP2* gene expression between circHIPK3 KD and scr samples in UMUC3 (p = 0.41, likelihood ratio test, *Figure 4—figure supplement 1H*) and only a slight, though significant, difference at later time points in J82 cells (p = 0.003, likelihood ratio test, *Figure 4—figure supplement 1H*), consistent with no direct effect of circHIPK3 on IGF2BP2 levels.

Taken together, our results suggest that circHIPK3 possesses important regulatory roles in BC by interacting with the oncogenic RBP IGF2BP2, which potentially affects proliferative pathways.

## circHIPK3 functions as a ceRNA for IGF2BP2

To further elucidate the functional role of circHIPK3–IGF2BP2 interactions in BC, we identified specific targets of circHIPK3 and IGF2BP2. An interesting example is the *STAT3* gene, which contains the 11-mer motif in its 3'UTR (allowing two mismatches), becomes downregulated upon circHIPK3 KD in UMUC3 (P_24h = 3e−04, Wald test, *Figure 5A*) and J82 cells (P_48h = 0.01, Wald test (*Figure 1—figure supplement 1*)), while it is upregulated upon IGF2BP2 depletion in both cell lines (p < 0.001, Wald test, *Figure 5B* and *Figure 5—figure supplement 1B*). IGF2BP2 RIP experiments confirmed a *STAT3*–IGF2BP2 interaction in BC cells (p = 0.04, *T*-test, *Figure 5C*). Interestingly, upon immunoprecipitation of IGF2BP2, we observed significantly more *STAT3* mRNA binding after circHIPK3 depletion (48 hr), supporting a regulatory role of circHIPK3 as a ceRNA (p < 0.007, *T*-test, *Figure 5D* and *Figure 5—figure supplement 1C*). We analyzed the IGF2BP2-binding affinity of three other transcripts (*NEU*, *SLC7A5*, and *TRAPPC9*) containing the 11-mer motifs in their 3'UTRs, which are also downregulated upon circHIPK3 KD, and conversely upregulated upon IGF2BP2 KD, and found a similar significant enrichment upon circHIPK3 KD (p < 0.03, *T*-test, *Figure 5—figure supplement 1D*). The relative IGF2BP2 binding of circHIPK3 (IP/INPUT ratio) was not affected by circHIPK3 KD, suggesting that excess IGF2BP2 does not affect the amount bound per circHIPK3 molecule (p = 0.35, *T*-test, *Figure 5—figure supplement 1E*).

If circHIPK3 acts as a ceRNA for IGF2BP2 and negatively affects target mRNA binding, we predict that the genes downregulated upon circHIPK3 KD, and conversely, upregulated by IGF2BP2 KD, becomes restored to normal levels if both circHIPK3 and IGF2BP2 are downregulated. To test this, we performed control (Scr), single (circHIPK3 or IGF2BP2) or double (circHIPK3 and IGF2BP2) KDs, followed by RT-qPCR on respective targets *STAT3*, *NEU*, and *TRAPPC9*. In support of our ceRNA model, the single KDs confirmed the previously observed expression pattern (down- and upregulation for circHIPK3 and IGF2BP2, respectively), while double KD indeed restored target mRNA expression in J82 cells (*Figure 5—figure supplement 1F*).

To assess the significance of the 11-mer motif in circHIPK3–IGF2BP2 interactions, we transfected FL3 cells with antisense oligonucleotides (ASO) specifically targeting the 11-mer motif in circHIPK3 but theoretically less so IGF2BP2 target genes, where up to two mismatches were allowed. Although not statistically significant, IGF2BP2-binding affinity to *STAT3* is 2.9 times higher in the presence of the 11-mer ASO compared to control samples (p = 0.078, *T*-test, *Figure 5E*). Additionally, the relative circHIPK3–IGF2BP2-binding affinity is 30% lower in 11-mer ASO samples (p = 0.27, *T*-test, *Figure 5—figure supplement 1G*). Next, to directly evaluate the importance of the 11-mer motif in IGF2BP2 binding of circHIPK3, we mutated the sequence in a pcDNA3.1+-Laccase2 circRNA expression construct and transfected HEK293 cells with either wildtype or mutant circHIPK3. Interestingly, we found that binding of IGF2BP2 was reduced significantly (~twofold, p = 0.028) when the 11-mer motif had been scrambled, while binding of *STAT3* mRNA to IGF2BP2 was increased (~sevenfold, p = 0.036) (*Figure 5F*). Importantly, cells expressed similar levels of RNase R-resistant WT and mutant circHIPK3 as judged by northern blotting (*Figure 5—figure supplement 1H*) and circHIPK3 was produced in ~fourfold excess of the linear precursor (*Figure 5—figure supplement 1I*).

It is well established that STAT3 regulates *TP53* expression by repressing its transcription (**Niu et al., 2005**). If the observed downregulation of *STAT3*, upon KD of circHIPK3 is functionally relevant, we predict that *TP53* mRNA and protein levels would rise. Consistent with our predictions, *TP53* mRNA is upregulated upon circHIPK3 KD at later time points in UMUC3 cells (p = 1.7e−05, Wald test, *Figure 5G*). Additionally, western blot analysis confirmed that circHIPK3 KD indeed leads to downregulation of STAT3 protein, while p53 protein levels rise (p < 0.05, *T*-test, *Figure 5H*). Finally, immunofluorescence staining showed that nuclear and overall STAT3 protein levels are significantly lower upon circHIPK3 KD (p < 0.003, Wilcoxon Rank Sum Test, *Figure 5I*), while nuclear p53 protein levels are higher (p = 1.7e−30, Wilcoxon Rank Sum Test, *Figure 5J*).

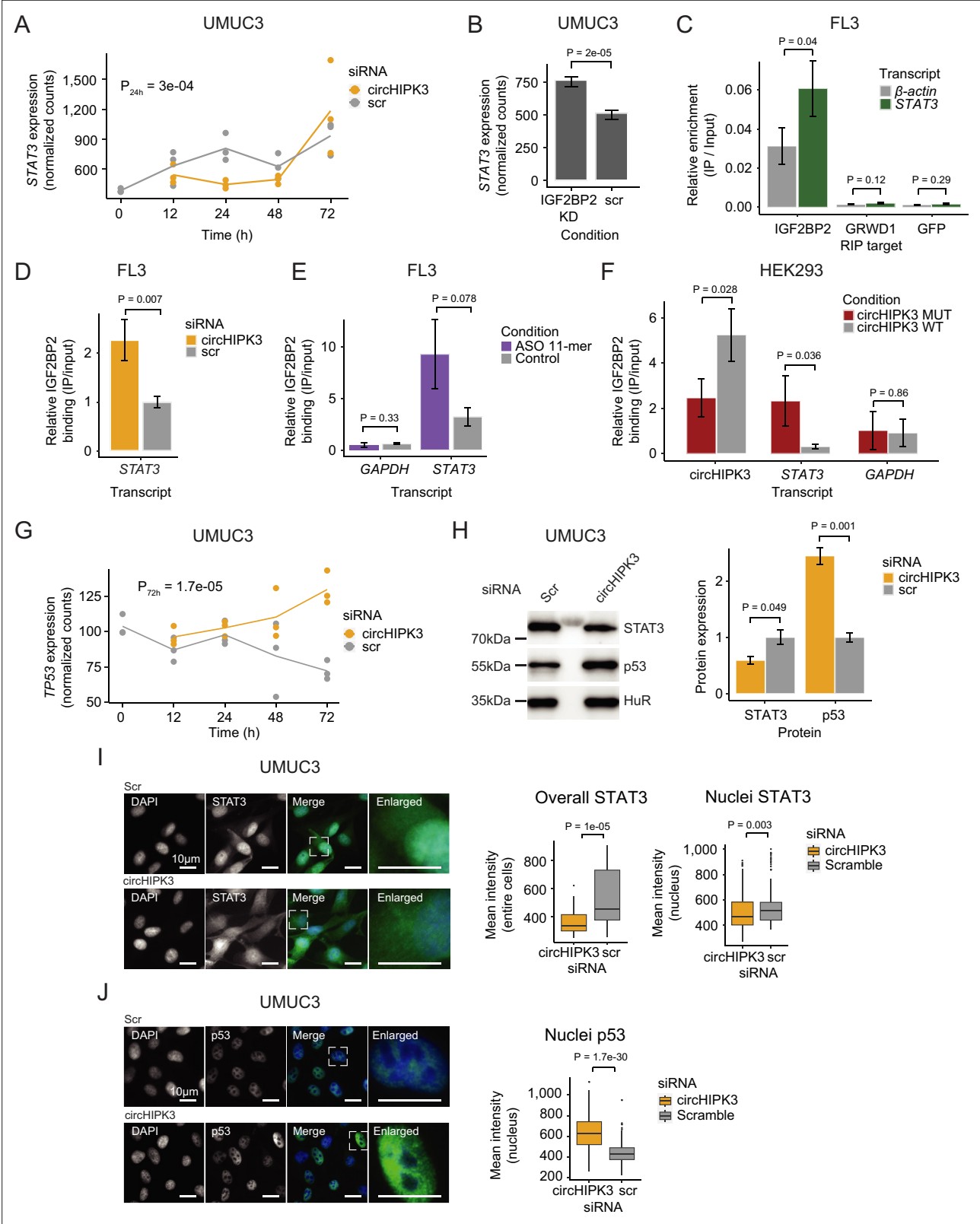

**Figure 5.** circHIPK3 functions as a competing endogenous RNA for IGF2BP2. (**A**) *STAT3* mRNA expression upon circHIPK3 knockdown (KD) in the time-course perturbation experiment in UMUC3 cells. Expression represents DESeq2 normalized counts. p-value (24 hr) obtained by Wald test. (**B**) Expression of *STAT3* upon IGF2BP2 KD in UMUC3 cells. p-value obtained by Wald test. (**C**) Relative enrichment of *STAT3* and β-actin levels between IGF2BP2 IP and input in FL3 cells. Green fluorescent protein (GFP) was used as a negative control. p-value obtained by *T*-test. (**D**) Relative enrichment of *STAT3*

*Figure 5 continued on next page*

*Figure 5 continued*

levels between IGF2BP2 IP and input upon circHIPK3 KD in FL3 cells. p-value obtained by *T*-test. Control sample (scr) has been normalized to 1. (**E**) Relative enrichment of *STAT3* and *GAPDH* levels between IGF2BP2 IP and input after 11-mer antisense oligonucleotide (ASO) transfection in FL3 cells. p-values obtained by *T*-test. (**F**) Relative enrichment of circHIPK3, *STAT3*, and *GAPDH* levels between IGF2BP2 IP and input after transfection of wildtype and mutated circHIPK3 in HEK293 cells. p-values obtained by *T*-test. (**G**) *TP53* mRNA expression upon circHIPK3 KD in the time-course perturbation experiment in UMUC3 cells. Expression represents DESeq2 normalized counts. p-value (72 hr) obtained by Wald test. (**H**) Western blot of STAT3 and p53 protein upon circHIPK3 KD in UMUC cells. Quantifications on the left are internally normalized to HuR and then scr is normalized to 1. p-values obtained by *T*-test. Immunofluorescence staining for STAT3 (**I**) or p53 (**J**) in fixed UMUC3 cells (as indicated – green) subjected to control (scr) or circHIPK3 KD. Nuclei were counterstained using 4',6-diamidino-2-phenylindole (DAPI) (as indicated – blue). Mean signal intensities are quantified within nuclei or entire cells to the right. p-values obtained by Wilcoxon Rank Sum Test. (**B–F, H**) Error bars reflect standard deviation of biological triplicates.

The online version of this article includes the following source data and figure supplement(s) for figure 5:

**Source data 1.** Original file for the western blot analysis shown in *Figure 5H* (anti-STAT3, anti-p53 and anti HuR).

**Figure supplement 1.** circHIPK3 functions as a competing endogenous RNA for IGF2BP2.

**Figure supplement 1—source data 1.** Original file for the northen blot analysis shown in *Figure 5—figure supplement 1* (backsplicing junction probe).

Our experiments suggest that circHIPK3 functions as a ceRNA for IGF2BP2 most likely by specifically interacting with IGF2BP2 at the 11-mer-binding site (and potentially secondary sites), which affects IGF2BP2-binding affinity to other target genes. This in turn represses STAT3 expression while upregulating p53.

## The ratio between circHIPK3 and IGF2BP2 cannot explain a simple ceRNA mode of action

To function as a ceRNA through direct binding, circHIPK3 levels must at least roughly match those of IGF2BP2. To directly assess the number of circHIPK3 and IGF2BP2 protein molecules per cell, we first designed a backsplicing junction DNA template for generation of qPCR standard curves (*Figure 6A* and *Figure 6—figure supplement 1A,B*). Isolating RNA from a known number of cells then allowed us to estimate the number of circHIPK3 molecules to ~45 (FL3), ~51 (J82), and ~40 (UMUC3) per cell (*Figure 6B* and *Figure 6—figure supplement 1C*). As IGF2BP2 is generally an abundant ubiquitous protein with an estimated ~10,000 copies per cell in HeLa cells (*Bekker-Jensen et al., 2017*), we next wanted to quantify IGF2BP2 content in FL3, J82, and UMUC3 cells. To this end, we expressed and purified Twin-streptagged-IGF2BP2 from HEK293 Flp-in T-Rex cells and quantified the yield by Coomassie staining by comparing to known amounts of bovine serum albumin (*Figure 6A* and *Figure 6—figure supplement 1D*). From known amounts of purified IGF2BP2 in western blots with lysates from a known number of BC cells, we estimate that each cell contains ~216,000 (FL3), ~127,500 (J82), and ~261,000 (UMUC3) IGF2BP2 molecules (*Figure 6B* and *Figure 6—figure supplement 1E, F*).

These results suggest that IGF2BP2 significantly outnumbers circHIPK3 by several orders of magnitude and hence that a ceRNA function for circHIPK3, in its simplest form, is unlikely. How can circHIPK3 then affect IGF2BP2 function? One possibility is that circHIPK3 nucleates numerous IGF2BP2 proteins on each circRNA molecule potentially starting at the 11-mer motif. To test this possibility, we performed immunofluorescence targeting endogenous IGF2BP2 in UMUC3 cells with circHIPK3 overexpression and hypothesized that if circHIPK3 nucleates IGF2BP2 molecules, it would be visible as cellular condensates. Indeed, we observed a significantly higher prevalence of large cytoplasmic condensates in cells expressing high levels of circHIPK3 compared to controls (80% vs 29%) (*Figure 6C, D*), which may contribute to IGF2BP2 accumulation by much fewer circHIPK3 molecules. Our results suggest that while circHIPK3 is unable to sequester IGF2BP2 in a 1:1 fashion, due to the large excess of IGF2BP2, the circRNA may efficiently nucleate numerous IGF2BP2 molecules, which are then growing into visible phase-separated condensates upon circHIPK3 overexpression.

## circHIPK3 expression is positively correlated with overall survival

Studies have shown that *HIPK3* possesses oncogenic features (*Curtin and Cotter, 2004*), while the derived circHIPK3 is downregulated in cancer and possesses tumor suppressor functions (*Teng et al., 2019*).

We extended a previously described cohort (*Hedegaard et al., 2016*; *Okholm et al., 2017*; *Strandgaard et al., 2022*; *Taber et al., 2020*), and analyzed the expression of circHIPK3 in more than

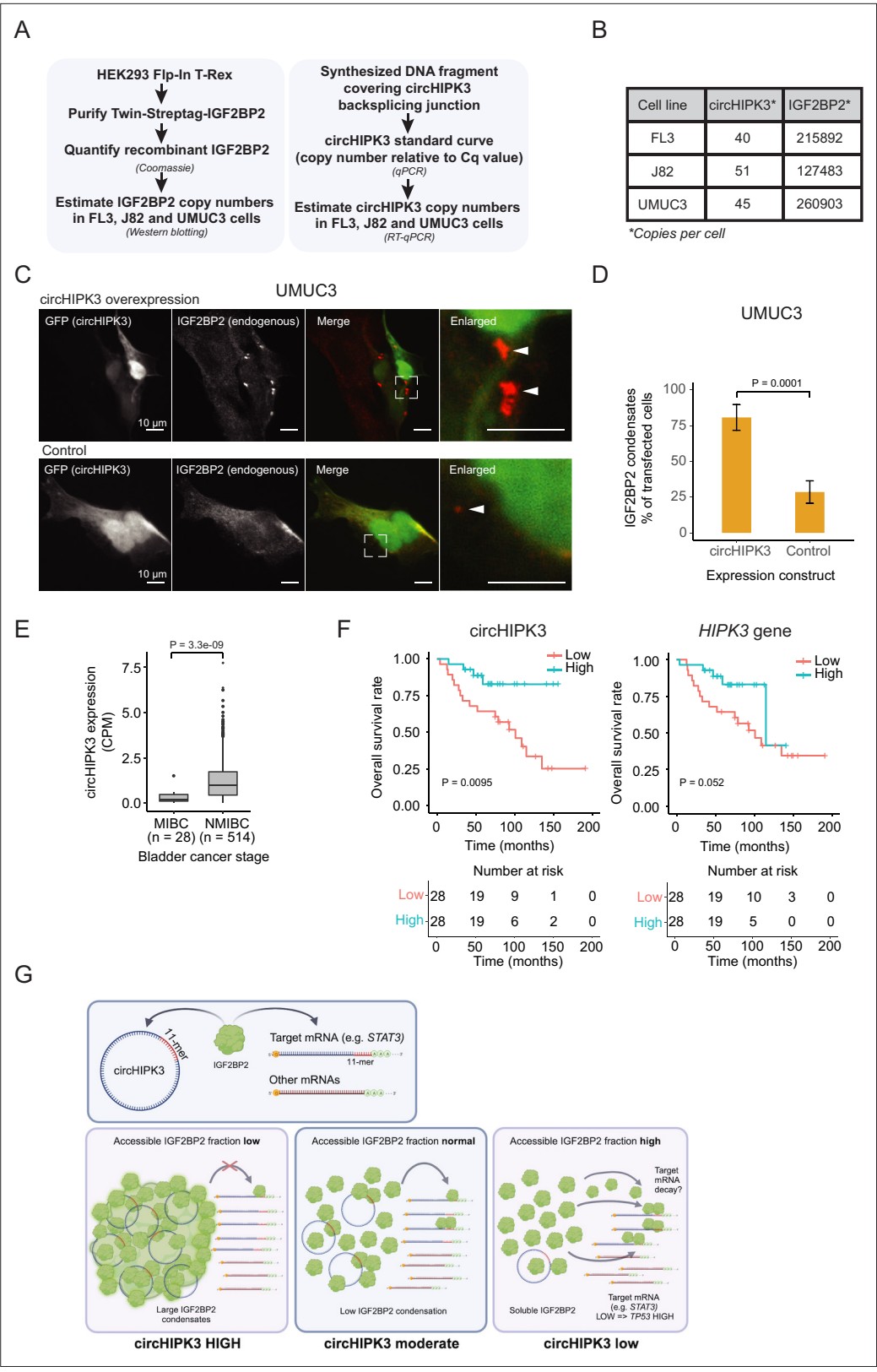

**Figure 6.** Absolute quantifications suggest low levels of circHIPK3 compared to IGF2BP2, where high circHIPK3 levels correlate with increased survival of bladder cancer (BC) patients. (**A**) Flow chart used to perform absolute quantification of circHIPK3 and IGF2BP2 protein levels in BC cells. (**B**) Estimated levels of circHIPK3 and IGF2BP2 (copies per cell). (**C**) Immunofluorescence using IGF2BP2 antibody (red channel) with cells transfected with

*Figure 6 continued on next page*

*Figure 6 continued*

circHIPK3 expression plasmid (upper panel) or control plasmid (lower panel). Both plasmids encode enhanced green fluorescent protein (EGFP) from a separate expression cassette in order to identify transfected cells (green channel). White arrows indicate IGF2BP2 condensates (**D**) Quantification of transfected cells containing large IGF2BP2 condensates – % positive cells (28 and 34 cells were counted). Error bars reflect standard error of mean (SE). (**E**) Expression of circHIPK3 in samples from patients with non-muscle invasive bladder cancer (NMIBC) and muscle invasive bladder cancer (MIBC). CPM = counts per million. p-value obtained by Wilcoxon Rank Sum Test. (**F**) Kaplan–Meier overall survival plots of circHIPK3 (left) and *HIPK3* (right). Median expression of circHIPK3 (0.192 CPM) and *HIPK3* (7.96 FPKM) used as cutoff. p-values obtained by Log-Rank Test. (**G**) Working model that illustrates potential mechanism for circHIPK3. Our studies show that circHIPK3 contains an 11-mer motif (illustrated by red bases) that comprise a binding site for IGF2BP2 (in green). At normal levels, circHIPK3 interacts strongly with IGF2BP2 (middle panel), potentially involving nucleation of many IGF2BP2 proteins on single circHIPK3 molecules – exacerbated at high circHIPK3 levels, which induce condensates (left panel). This allows for only a few accessible IGF2BP2 molecules to bind target mRNAs including *STAT3*. Upon circHIPK3 knockdown (KD) and low circHIPK3 levels (right panel), IGF2BP2 is more accessible to interact more strongly with IGF2BP2 target genes, which contain the 11-mer motif in their 3'UTRs, incl. *STAT3*, and which are subsequently downregulated. Conversely, upon IGF2BP2 KD, IGF2BP2 target genes containing 11-mer are upregulated (not illustrated).

The online version of this article includes the following source data and figure supplement(s) for figure 6:

**Figure supplement 1.** Absolute quantification of IGF2BP2 and circHIPK3 in bladder cancer cells.

**Figure supplement 1—source data 1.** Original file for the scanned Coomassie brilliant blue stained gel shown in *Figure 6—figure supplement 1*.

**Figure supplement 1—source data 2.** Original file for the western blot analysis shown in *Figure 6—figure supplement 1* (anti-IGF2BP2).

500 samples from patients with BC. The expression of circHIPK3 is significantly upregulated in patients with NMIBC compared to patients with muscle invasive BC (p < 3.3e−09, Wilcoxon Rank Sum Test, *Figure 6E*).

We previously reported that circHIPK3 expression correlates positively with BC progression-free survival independently of the parent gene (*Okholm et al., 2017*). In a newly generated and hence independent cohort of patients with NMIBC (*n* = 56) (*Strandgaard et al., 2022*), we found that circHIPK3 levels are positively associated with overall survival (p < 0.009, Log-Rank Test, *Figure 6F*). The association was less pronounced for *HIPK3* levels (p = 0.052, Log-Rank Test, *Figure 6F*). Our clinical analyses confirm that high circHIPK3 expression levels are correlated with good clinical outcomes.

In conclusion, our study provides evidence that the most prominent alpha circRNA, circHIPK3, interacts with IGF2BP2 to regulate the expression of hundreds of genes containing the 11-mer motif in its 3'-UTR, including *STAT3*. Since circHIPK3 expression induces IGF2BP2 condensates, we suggest that such protein–RNA nucleation events may, also at the submicroscopic level, enhance the potential for circHIPK3 to titrate out multiple IGF2BP2 molecules even though it is found in low numbers within cells (*Figure 6G*). Alternatively, circHIPK3 may recruit unknown enzymatic modifiers of IGF2BP2, which in turn could affect its activity. Both scenarios could explain the functional output from a small number of circHIPK3 molecules in cells and await further experimentation.

## Discussion

In this study, we characterized the functional role of circHIPK3 in BC cells by perturbation studies, comprehensive analyses of large genomics datasets, and experimental hypothesis evaluation. In contrast to previous studies suggesting a role for circHIPK3 as a miRNA sponge (reviewed in *Zhou et al., 2021*), our results indicate that manipulation of circHIPK3 levels does not affect genes targeted by miRNAs with seed region-binding sites also present within circHIPK3. Specifically, based on AGO-CLIP data profiling miRNA target sites from 32 independent experiments, we found no indication that circHIPK3 functions as a miRNA sponge. In addition, since most reports claiming circRNA-specific miRNA sponging are based on inflated circRNA expression, use of miRNA mimics and reporter assays, a conclusion that circRNAs are generally functioning as miRNA sponges should be taken with some caution. Indeed, our estimation of less than a hundred circHIPK3 molecules per cell, questions the miRNA-sponge hypothesis, as this entails specific sequence-specific binding of miRNA-guided Argonaute proteins. It should be noted however, that miRNA expression and target sites are often tissue

specific and the lack of BC tissue in the AGO-CLIP data could influence our findings. Nonetheless, several studies argue against a general miRNA-sponge function for most circRNAs (*Guo et al., 2014*; *Militello et al., 2017*).

Here, we identified an 11-mer motif present in the sequence of circHIPK3 that constitutes a binding site for IGF2BP2 in RIP assays. IGF2BP2 is a multifunctional RBP that affects hundreds of mRNAs and various RNA processes (*Cao et al., 2018*). It belongs to a family of three conserved RBPs (IGF2BP1, IGF2BP2, and IGF2BP3), that are important modulators of gene expression, especially during development, and contain highly similar RNA-binding domains; two N-terminal RNA-recognition motifs and four C-terminal hnRNPK Homology (KH) domains (reviewed in *Bell et al., 2013*). Recently, Biswas identified a consensus binding motif for IGF2BP2, CUCAC-(N10-15)-(A/U)-GG-(A/U), containing the core elements CA and GG required for specific binding of the IGF2BP2 KH3 and KH4 domain, respectively (*Biswas et al., 2019*). Although in reverse order, both core REs are present in the 11-mer motif, and we observed a significant overlap between genes containing the consensus sequence and the 11-mer motif (allowing two base substitutions).

While some RBPs contain well-established binding domains and bind to well-defined motifs, most RBPs rely on sequence context and structural features (*Dominguez et al., 2018*; *Doyle and Kiebler, 2012*), suggesting that RNA structure could explain the discrepancies observed here. Additionally, IGF2BP2 might recognize the swapped orientation of the REs, as observed for IGF2BP1 (*Patel et al., 2012*). Although earlier work has shown that the KH34 domains are the major RNA-binding domains of all IGF2BPs (*Chao et al., 2010*; *Patel et al., 2012*), more recent studies demonstrate that additional RNA-binding domains are involved in RNA recognition and distinction between targets (*Schneider et al., 2019*; *Wächter et al., 2013*), complicating the establishment of a consensus motif and binding sites predictions. Furthermore, other external factors likely influence binding affinity and preference, and cooperativity both intra- and intermolecularly might be restricted to certain cells, tissues, and environment, as observed for circFoxo3, which interacts with different RBPs depending on the biological setting (*Du et al., 2017b*; *Du et al., 2017a*; *Du et al., 2016*). Since nucleotide sequence is thus insufficient to fully predict RBP-binding sites, we used large-scale motif-enrichment analysis of RNA-Seq data and experimentally validated RBP target sites from eCLIP data to identify circHIPK3–IGF2BP2 mRNA targets and regulatory tendencies that are consistent across ENCODE and BC cell lines. While we do not attempt to define a new recognition motif for IGF2BP2, our results are in line with previous work describing the necessity of two appropriately spaced REs that are crucial for binding by all IGF2BPs (*Biswas et al., 2019*; *Patel et al., 2012*; *Schneider et al., 2019*).

While circHIPK3 is frequently downregulated in cancer and positively correlated to good clinical outcomes (*Li et al., 2017*; *Okholm et al., 2017*), IGF2BP2 enhances genomic instability, is upregulated in cancer, and involved in cancer progression (*Barghash et al., 2016*; *Kessler et al., 2015*; *Ye et al., 2016*). Our results show that IGF2BP2 KD counteracts the effect of circHIPK3 KD on target genes and proliferation pathways, indicating that circHIPK3 might function as a reservoir for IGF2BP2. By focusing on circHIPK3 and IGF2BP2 regulated targets, we provide evidence that circHIPK3–IGF2BP2 interactions affect the expression of *STAT3*. STAT3 is persistently activated in a wide diversity of cancers and is positively associated with cell proliferation, cell survival, invasion, and metastasis (reviewed in *Kamran et al., 2013*), although tumor suppressor activities under certain conditions, and in certain cancers, have also been reported for STAT3 (reviewed in *Tolomeo and Cascio, 2021*). At this point, we cannot explain why downregulation of a proposed tumor suppressor circRNA (*Okholm et al., 2017*) also downregulates a STAT3, whose activation is most often associated with cancer progression. Indeed, STAT3 phenotypes are multifaceted making correlations between short- and long-term downregulation of circHIPK3 very complex. By conducting RIP experiments after circHIPK3 KD, we observed a stronger IGF2BP2 association with *STAT3* mRNA (~sevenfold enrichment), along with a concomitant *STAT3* mRNA and protein downregulation. Consistent with this finding, a study recently used an mRNA–RBP tethering reporter assay and found a general mRNA destabilizing effect of an IGF2BP2–mRNA association (*Luo et al., 2020*). This strong association is dependent on the 11-mer sequence, since a circHIPK3 variant with a mutated motif decreased binding significantly in RIP assays. These results point to a regulatory role of circHIPK3 as a ceRNA for IGF2BP2 binding.

Surprisingly, our stoichiometric analysis of copy numbers of circHIPK3 and IGF2BP2 in BC cell lines are not readily compatible with a simple ceRNA hypothesis, since IGF2BP2 outnumbers circHIPK3 by orders of magnitude. However, we demonstrated that exogenous expression of circHIPK3 produces

IGF2BP2-containing condensates, suggesting that circHIPK3 may initiate a nucleation process that will likely condensate/phase separate multiple proteins, which could at least partly explain how a few cellular circHIPK3 molecules can sequester many IGF2BP2 proteins. At this point, we cannot exclude alternative mechanisms, including that the cell only has a limited number of available IGF2BP2 molecules that can dynamically interact with either circHIPK3 or other mRNA targets, leaving enough circles for a proper ceRNA function. More speculatively, circHIPK3 may scaffold an enzymatic complex that can modify IGF2BP2 post-translationally and change its function (e.g., RNA-binding, localization or interaction with other factors) – a scenario where a few circRNAs can change the function of numerous RBPs. Still, the regulatory interplay could be more complex and circHIPK3 could act as a scaffold that facilitates interaction between several RBPs or as part of a larger functional unit. While we provide evidence that circHIPK3 interacts with IGF2BP2 to regulate IGF2BP2 target genes, the detailed molecular mechanism remains to be fully elucidated.

Taken together, we identified an 11-mer motif within circHIPK3 that is important for the regulation of multiple cancer-related genes via IGF2BP2 binding. We propose a ceRNA model where a few circHIPK3 molecules can sequester multiple IGF2BP2 proteins by inducing their condensation.

## Materials and methods
### Plasmids
To generate plasmids for exogenous expression of Twin-Strep-tagged IGF2BP1 (transcript variant 1, NM_006546.4), GRWD1 (transcript variant 1, NM_031485.4) or green fluorescent protein (GFP), cDNA libraries from HEK293 and HeLa cells were used as templates for standard PCR reactions using primers containing BamHI–NotI restrictions sites that amplify the CDS. XhoI–NotI restriction sites were used for cloning IGF2BP2 (transript variant 1, NM_006548.6). The amplicons were inserted into pcDNA5-Twin-Streptag. pcDNA5-Twin-Streptag was made by inserting a PCR product (containing a Twin-Streptag followed by a TEV protease site) using a pDSG-IBA-Twin-Strep-Tag vector as template (IBA-lifesciences) into the HindIII–ApaI site of pcDNA5-FRT-TO (Invitrogen).

To generate plasmids for exogenous expression of circHIPK3, the second exon encoding circHIPK3 was PCR amplified from cDNA prepared from RNA isolated from HEK293/HeLa cells and inserted into PacI/SacII-digested pcDNA3.1(+)-Laccase2-MCS-exon-vector (*Kramer et al., 2015*) to generate the pcDNA3.1(+)-Laccase2-circHIPK3 vector. pcDNA3.1(+) Laccase2-MCS-Exon-Vector was a gift from Jeremy Wilusz (Addgene plasmid # 69893; https://www.addgene.org/69893/; RRID:Addgene_69893). For overexpression of circHIPK3 followed by IGF2BP2 immunofluorescence, we modified the pcDNA3.1(+)-Laccase2-circHIPK3 vector to include an additional enhanced green fluorescent protein (EGFP) cassette to allow for screening of transfected cells. This was done by inserting an EGFP PCR fragment into the AvrII–BssHII sites of the vector thereby disrupting/replacing the NeoR resistance gene. The 11-mer motif mutation within pcDNA3.1(+)-Laccase2-circHIPK3 vector was generated by overlap-extension PCR including overlapping primers containing the mutated 11-mer sequence.

> HIPK3 Scr 11-mer F: TTGCATTTCCTAGAAGACGGCTACAGATGTGGATTGAAGCGCAAGAGTG
>
> HIPK3 Scr 11-mer R: CTTCAATCCACATCTGTAGCCGTCTTCTAGGAAATGCAATCTGTTTCGC

Outer amplification primers: Laccase F: CAACGAATCTAGTATACCTT. Laccase R: ACAATAGTATAG AAACTGCC. All constructs were verified by sanger sequencing.

### Cell lines – maintenance and growth conditions
Human BC cell lines UMUC3 (American Type Culture Collection (ATCC), CRL-1749), J82 (American Type Culture Collection (ATCC), HTB-1), FL3 (Cellosaurus T24T FL3 (CVCL_M897)) and human embryonic kidney cell line HEK293 Flp-In T-Rex cells (Cellosaurus Flp-In-T-REx-293 (CVCL_U427)) were maintained in Dulbecco's modified Eagle medium (DMEM) (Gibco, Thermo Fisher Scientific) including 10% fetal bovine serum (FBS; Gibco, Thermo Fisher Scientific) and penicillin–streptomycin (Gibco, Thermo Fisher Scientific) at a final concentration of 50 I.U./ml penicillin unless otherwise stated. UMUC3 and J82 cells were used in the time-course KD followed by RNA-Seq and in additional follow-up experiments, while FL3 was mainly used for circHIPK3 KD/overexpression and immunoprecipitation experiments primarily due to more efficient plasmid transfections. FL3 cells were also subjected to control/

circHIPK3 and RNA-Seq at 48 hr post-transfection and similar expression profiles were obtained between the three cell lines suggesting that these can all be used for functional studies of circHIPK3. K562 cells (American Type Culture Collection (ATCC), CCL-243) were grown in RPMI 1640 media (Gibco, Life Technologies, 11875119) with 10% FBS (Gibco Life Technologies, 26140079). HepG2 cells (American Type Culture Collection (ATCC), HB-8065) were grown in DMEM (Gibco, Thermo Fisher Scientific) with 10% FBS (Gibco Life Technologies, 26140079).

## circHIPK3 expression in cell lines

Whole transcriptome RNA-Seq data from 11 BC cell lines (HCV29, HT1197, Nhutert, RT4, FL3, HT-1376, J82, LUL2, SLT4, T24, and UMUC3) were generated and mapped in a previous study (*Hedegaard et al., 2016*). We used the CIRI2 pipeline (v2.0.6) (*Gao et al., 2018*) to profile the expression of circHIPK3. Before running the CIRI2 pipeline, we trimmed reads with trim_galore (v0.4.1). We aligned the reads to the human genome (hg19) using bwa (v0.7.15) and samtools (v1.3). The CIRI2 pipeline was run with a GTF file (hg19) to annotate the overlapping gene of the circRNAs.

For expression of circHIPK3 in HepG2 and K562 cells, we downloaded all total and fractionated samples for the cell lines HepG2 (*n* = 5) and K562 (*n* = 11) from ENCODE (https://www.encodeproject.org/) as described in *Okholm et al., 2020*. CircHIPK3 expression was profiled using the CIRI2 pipeline as described above for the BC cell lines and in *Okholm et al., 2020*.

## siRNA-mediated KD and RNA sequencing

A total of $3 \times 10^5$ cells (UMUC3 or J82) were seeded per well in 6-well plates (2 ml per well). Twenty-four hours later, cells were transfected with an siRNA targeting the unique backsplice junction of circHIPK3 (sense: GGUACUACAG|GUAUGGCCU[dTdT] and antisense: AGGCCAUAC|CUGUAGUACC[dTdT], "|" denote backsplice junction) using SilentFect (Bio-Rad) in biological triplicates: 100 µl OptiMEM (Gibco Life Technologies) + 2 µl SilentFect (Bio-Rad) was incubated at room temperature (RT) for 5 min before mixing with 2 µl siRNA (20 µM stock) pre-diluted in 100 µl OptiMEM. After gentle mixing by pipetting, the solution was incubated for 20 min at RT prior to dropwise addition to cells (final concentration of 20 nM). Cells were incubated for 12, 24, 48, or 72 hr before harvest by addition of 1 ml Trizol (Thermo Fisher Scientific) to each well followed by mixing. As negative control, we used MISSION siRNA Universal Negative Control #2 (Sigma-Aldrich). For KD of mRNAs encoding RBPs, UMUC3, J82, or FL3 cells were transfected as indicated above and incubated for 48 hr. Harvested RNA was purified according to the manufacturer's protocol with an additional 400 µl chloroform extraction to increase RNA quality. A parallel set of identically treated samples were used to harvest cells for protein lysates by addition of 300 µl 2× sodium dodecyl sulfate (SDS) sample buffer [4% SDS, 20% glycerol, 10% 2-mercaptoethanol, 0.004% bromophenol blue, and 0.125 M Tris–HCl, pH 6.8] per well followed by incubation at 90°C for 8–10 min until all cell material is dissolved.

KD efficiency of circHIPK3 was assessed by qRT-PCR. RNA samples were treated with DNase I (Thermo Fisher Scientific) according to the manufacturer's protocol. First-strand cDNA synthesis was carried out using the Maxima First Strand cDNA synthesis Kit for qPCR (Thermo Fisher Scientific) according to the manufacturer's protocol. qPCR reactions were prepared using gene-specific primers and Platinum SYBR Green qPCR Supermix-UDG (Thermo Fisher Scientific) according to the manufacturer's protocol. An AriaMx Real-time PCR System (Agilent Technologies) was used for quantification of RNA levels and the $X_0$ method was used for calculations of relative RNA levels (*Thomsen et al., 2010*) normalized to GAPDH or beta-actin (*ACTB*) mRNA as indicated.

We quantified mRNA expression using QuantSeq (*Moll et al., 2014*). Initial quality control was performed using Dropsense96 (Trinean) and 2100 Bioanalyzer (Agilent). Sequencing libraries were generated using the QuantSeq 3'mRNA Library Prep Kit Protocol (Lexogen). We used 1000 ng RNA as input and 11 PCR cycles were applied. Library sizes were measured on LabChip GX (PerkinElmer) or Bioanalyzer (Agilent) while concentrations were measured on Qubit 3.0 (invitrogen). Finally, we applied the KAPA Library Quantification Kit (Kapa Biosystems) to ensure library quality for sequencing.

The raw reads were converted to fastq format and demultiplexed using Illumina's bcl2fastq v2.20.0.422 and library adapters were removed from the read pairs (trim_galore v0.4.1). Reads were mapped to the human genome (hg19) using TopHat2 (version 2.1.1) (*Kim et al., 2013*) and Bowtie2 (version 2.1.0.0) (*Langmead and Salzberg, 2012*), and Cufflinks (v2.1.1) (*Trapnell et al., 2010*) and HTSeq (v0.6.1p1) (*Anders et al., 2015*) were used to estimate the transcripts abundance using

transcript information from GENCODE v19. Samtools (v1.3) (*Li et al., 2009*) and Picard (v2.0.1) were used for quality control and statistics.

## Differential expression analysis of time-series experiments

For circHIPK3 KD experiments, each cell line was analyzed separately. For the differential expression analysis, we excluded protein coding genes with a maximum expression <1 RPM and the 25% lowest-variance genes across samples. We used DESeq2 on raw counts to identify genes that are DE between conditions (circHIPK3 and scr siRNAs) or time points (*Love et al., 2014*). DESeq2 output files are available as *Supplementary files 6 and 7*. For each time point, we used the Wald test as implemented in DESeq2 to test for differential expression between conditions. Within conditions, we similarly applied the Wald test between pairs of time points. We also used DESeq2 to detect genes where the relative expression profiles across the time-series differ between the two conditions. Briefly, in DESeq2, this is achieved by testing for a significant interaction between condition and time using a likelihood ratio test. The FDR was controlled using Benjamini–Hochberg correction and significance was declared at FDR <0.1.

## miRNA expression profiling by NanoString nCounter

The nCounter Human v3 miRNA panel (NanoString Technologies), which targets 799 miRNAs, was used for miRNA profiling in the cell line J82 according to the manufacturer's instructions. In brief, 100 ng of total RNA input and a hybridization time of 20 hr was used. The products were analyzed on the nCounter SPRINT (NanoString Technologies) platform. The raw data were processed using the nSOLVER 4.0 software (NanoString Technologies); first, a background subtraction was performed using the max of negative controls, and then positive control normalization was performed using the geometric mean of all positive controls. Finally, a second normalization was performed using the expression levels of *GAPDH*.

## AGO-CLIP data of miRNA target sites

Generation of AGO-CLIP data was described previously (*Hamilton et al., 2016*; *Hamilton et al., 2013*; *Okholm et al., 2017*). To identify miRNA targets in the sequence of circHIPK3, we restricted our analysis to conserved miRNAs as this reduces false positives as described in *Okholm et al., 2017*. We evaluated the presence of miR-1207-5p in the entire AGO-CLIP data with no restriction on conservation.

## Identification of motifs in deregulated genes

We used Regmex (*Nielsen et al., 2018*) to identify motifs enriched in 3'UTRs of down- and upregulated genes. For circHIPK3 KD experiments, we ranked genes according to a differential expression score calculated as the −log10(p-value) that is signed according to whether the genes were upregulated (positive) or downregulated (negative). The score was calculated on gene regulation 24 hr post-transfection. Regmex was run with all 1032 unique 7-mers found in circHIPK3.

For ENCODE IGF2BP2 KD experiments, genes were ranked by the fold change expression between KD and control samples, and Regmex was run with all possible 7-mers.

## eCLIP of HepG2 and K562

The eCLIP data of RBP targets from HepG2 and K562 were obtained from *Van Nostrand et al., 2016* and processed as described in a previous study (*Okholm et al., 2020*). Briefly, we only considered RBP-binding sites that are at least eightfold enriched in the RBP immunoprecipitation under consideration compared to a size-matched input control. We restricted our analysis to RBPs with binding sites in exonic regions from known and novel protein-coding transcripts (hg19). RBP-binding sites <4 bp or an exon-RBP-binding site overlap <4 bp were disregarded. When analyzing enrichment of RBP-binding sites in genes affected by circHIPK3 KD, only the genes expressed in the ENCODE cell line and the BC cell line under consideration were evaluated.

## RNA immunoprecipitation

The RIP experiments in HepG2 and K562 were performed as described in *Okholm et al., 2020*. The following antibodies were obtained for RIP in HepG2 and K562 cells: GRDW1 (Bethyl A301-576A Lot 1), UCHL5 (Bethyl A304-099A Lot 1), YBX3 (A303-070A Lot 1), IGF2BP1 (MBL RN007P Lot 004), and

IGF2BP2 (MBL RN008P Lot 005). Rabbit IgG Isotype Control (Invitrogen Cat# 02-6102) was used as negative control. $20 \times 10^6$ snap frozen cells were lysed in 1 ml of iCLIP Lysis Buffer (50 mM Tris–HCl pH 7.4, 100 mM NaCl, 1% NP-40 (Igepal CA630), 0.1% SDS, 0.5% sodium Deoxycholate) with 5.5 µl Protease Inhibitor Cocktail Set III EDTA Free (EMD Millipore Corp.539134-1ML) and 11 µl Murine RNase Inhibitor (New England BioLabs Inc M0314L) for 15 min and were then centrifuged at 20,000 $\times$ $g$ for 20 min at 4°C. The supernatant was placed in a solution containing specific primary(10 µg)/ secondary(1.25 g) (anti-Rabbit magnetic DynaBeads, Invitrogen, 11204) antibody–antibody (incubated on a rotator at 25°C for 45 min) to immunoprecipitate overnight on a rotator at 4°C. The RNA–RBP pull-down was then purified by stringently washing with NET-2 wash buffer (5 mM Tris–HCl pH 7.5, 150 mM NaCl, 0.1% Triton X-100). Isolation of RNA from the RNA–RBP complexes was accomplished with the addition of TRIzol Reagent (Invitrogen, 15596018) followed by the Direct-zol RNA MiniPrep (Zymo Research Cat No. R2052). Isolated RNA was reverse transcribed with SuperScript III First Strand Synthesis System (Invitrogen 18080051) using Random Hexamers (Thermo Fisher Scientific N8080127) and circRNAs were semi-quantitatively amplified using GoTaq DNA Polymerase (Promega, M3005); 4 µl of 1:2.5 diluted cDNA, 1 µl of each primer for 34 cycles at the following conditions: strand separation at 95°C for 30 s, primer hybridization at 55°C for 30 s, and elongation at 72°C for 20 s followed by a final elongation step at 72°C for 5 min. Amplicons were run on a 3% Agarose Gel at 135 V for 35 min at 4°C alongside a 50-bp ladder marker. CircHIPK3 bound by RBPs was identified on the gel based on amplicon size.

For RIP in BC cells, $1.65 \times 10^6$ FL3 cells were seeded in a P10 dish and transfected 24 hr later with 12 µg total DNA (2 µg Twin-Strep-Tag-RBP expression vector and 10 µg pcDNA3 PL) using Lipofectamine 2000 (Invitrogen) according to the manufacturer's recommendations. Forty-eight hours later cells were washed with phosphate-buffered saline (PBS) and placed on ice. One milliliter cold lysis buffer (50 mM Tris–HCl pH 7.5, 10 mM NaCl, 5 mM MgCl$_2$, 0.5% Triton X-100, 1% Hexane-1,6-diol, 1 pill Complete – protease inhibitor ethylenediaminetetraacetic acid EDTA-free (Roche) pr. 10 ml) was added per plate and cells were scraped off and transferred to an Eppendorf tube. Samples were mixed and spun (13,000 RPM, 15 min at 4°C) and 50 µl supernatant was transferred to 50 µl SDS load buffer while 100 µl supernatant was transferred to 0.5 ml Trizol (INPUT). 800 µl supernatant was incubated with pre-equilibrated MagStrep type 3 XT beads (IBA Life sciences) and rotated for ≥2 hr at 4°C. 500 µl supernatant was collected (FT [flow-through]) and samples were washed 4× with 1.5 ml WASH1 buffer (10 mM Tris–HCl pH 7.5, 150 mM NaCl, 5 mM MgCl$_2$, 0.1% Triton X-100) and 2× with WASH2 buffer (10 mM Tris–HCl pH 7.5, 150 mM NaCl, 5 mM MgCl$_2$). During the last wash sample beads were divided 1:1 (RNA:protein) and added 50 µl SDS load buffer or 0.5 ml Trizol (IP). The incubation time for siRNA KD was 48 hr. INPUT, FT, and IP samples were analyzed with western blotting and RT-qPCR (as described above) for indicated targets, including circHIPK3 or low-affinity control *ACTB* mRNA.

## Western blotting

Western blotting was performed by standard procedures using Novex Tris-Glycine buffered gradient WedgeWell gels (8–16%) (Invitrogen – Thermo Fisher Scientific) in a Mini Cell apparatus. All protein transfers were performed overnight onto Amersham Hybond (Cytiva) polyvinylidene difluoride (PVDF) membranes 0.45 µm pore size. Chemiluminescence from horse radish peroxidase (HRP)-conjugated secondary antibodies (1:20,000 dilution) (Thermo Fisher Scientific) was developed using SuperSignal, West Femto Maximum Sensitivity Substrate (Thermo Fisher Scientific). Antibodies used: Mouse anti-Streptag II antibody (Antibodies Online) ABIN7250959 (1:1000), Mouse anti-STAT3 (Cell Signaling Technologies, CST-9139) (1:1000), Mouse anti-HuR (Santa Cruz Biotechnology) (SC-365338) (1:5000), Rabbit anti-p53 (Cell Signaling Technologies, CST-2527) (1:500), and Rabbit anti-HIPK3 (Abcam Ab72538) (1:500). Acquisition and quantifications of band intensities were performed using a Licor OdysseyFc scanner and Image Studio version 5.2 software. Band intensities were normalized to abundant house-keeping protein HuR or in some cases total protein assessed by Revert Stain (Licor). The normalized band intensities are based on three biological replicates (quantified western blots – plotted data). Error bars represents standard deviations from means ($n$ = 3).

## Northern blotting

Northern blotting was carried out largely as previously described in *Hollensen et al., 2020*. In brief, for each sample 2 × 15 µg RNA were aliquoted into two separate Eppendorf tubes. One half of the tubes

were subjected to RNase R-treatment. The RNA was digested with 10 units of RNase R (Biosearch Technologies) for 20 min at 37°C. All RNA samples were ethanol precipitated overnight and resuspended in 1:1 formamide load buffer (deionized formamide, 5 mM EDTA) and 2× 3-(N-morpholino)propanesulfonic acid (MOPS) load buffer (2× MOPS, 11% formaldehyde, 5 mM EDTA, 0.18% xylene cyanol/bromophenol blue). The RNA was then separated in a 1.2% formaldehyde-agarose gel, followed by transfer to a Hybond-N+ membrane (Cytiva). To visualize ribosomal RNA, the membrane was stained with a 0.02% Methylene Blue/3 M NaAc solution. Subsequently, the membrane was hybridized with Laccase-circHIPK3- or 5.8S-specific [32P]-end-labeled 40-mer oligonucleotides overnight at 42°C in ULTRAhyb-Oligo hybridization buffer (Ambion), exposed on a phosphorimager screen for several days, and finally visualized on a Typhoon FLA 9500 (GE Healthcare).

## IGF2BP2 KD

RBP KD samples in K562 and HepG2 were produced for the ENCODE project by Brenton Graveley's Lab, UConn (*The ENCODE Project Consortium, 2012*). RNA-Seq data of polyadenylated transcripts were obtained from ENCODE (https://www.encodeproject.org/). When comparing genes affected by IGF2BP2 KD and circHIPK3 KD, only the genes expressed in the ENCODE cell line KD experiment and the BC cell line under consideration were evaluated. IGF2BP2 KD in the BC cell lines UMUC3 and J82 were described in a previous study (*Okholm et al., 2020*).

## Pathway analysis

We used the R package gage (*Luo et al., 2009*) for pathway analyses of 50 hallmarks of cancer gene sets from The Molecular Signatures Database (MSigDB) (*Liberzon et al., 2015*). For exploratory pathway analyses upon circHIPK3 KD in UMUC3, we used a significance level of FDR <0.1. To verify specific observations upon circHIPK3 KD in J82 and IGF2BP2 KD in UMUC3 and J82, we used a significance level of $p < 0.05$.

## EdU Click-IT proliferation assay

The cell cycle assay was performed as described in detail elsewhere (*Hollensen et al., 2020*). Briefly, labeling of newly synthesized DNA was carried out using Click-iT Plus EdU Alexa Flour Flow Cytometry Assay Kit (Thermo Scientific) according to the manufacturer's protocol. Notably, the cell culture medium of FL3 cells cultured in 6 cm dishes was supplemented with 10 µM EdU for 90 min. To stain total DNA, cells with already detected EdU were resuspended in 300 µl 1× ClickIT saponin-based permeabilization and wash reagent from the assay kit. Subsequently, RNase A was added to a final concentration of 0.2 mg/ml. After 5 min of incubation at RT, propidium iodide was added to a final concentration of 5 µg/ml and the cells were incubated for 30 min at RT. Incorporated EdU and total DNA levels were analyzed on a NovoCyte Quanteon Flow Cytometer System (Agilent). The incubation time for siRNA KD was 48 hr. Data analysis was carried out in the FLOWJO software (BD Biosciences).

## Immunofluorescence microscopy

Autoclaved 18 mm coverslips were placed in 12-well plates and covered with 0.01% poly-L-lysine for 2–5 min and allowed to dry. The dry poly-L-lysine coated coverslips were then placed under UV light for 10 min before sealing with Parafilm. The coated coverslips were stored at RT until use. siRNA-transfected cells grown on poly-L-lysine coated coverslips for 48 hr and washed twice in PBS and fixed using 4% paraformaldehyde in PBS at RT for 15 min. When performing overexpression of circHIPK3 using pcDNA The fixed cells were then washed 3× in 1 ml PBS. Fixed cells were permeabilized using 1% horse serum/0.5% Triton X-100 in PBS at RT for 10–15 min and then washed twice in PBS containing 1% horse serum. Cells were incubated with primary antibodies diluted (1:500–1:1000) in PBS/1% horse serum for 1 hr, prior to 3 × 1 ml washes in PBS/1% horse serum. Secondary antibodies AlexaFlour 488 (Thermo Scientific) was used at a dilution of 1:1000 in PBS/1% horse serum and incubated 1 hr at RT prior to 3× washes in PBS/1% horse serum. Cells were then counterstained with 1 µg/ml DAPI in PBS/1% horse serum for 15–30 s before washing with PBS. Coverslips were rinsed in ddH$_2$O and mounted onto a microscope slide using ProLong Gold Antifade Mountant. Antibodies: Rabbit anti-p53, Cell Signaling Technologies (CST), 2527 and Mouse anti-STAT3, Cell Signaling Technologies (CST), 124H6, 9139 and Rabbit anti-IGF2BP2 Cell Signaling Technologies (CST). Images were acquired at ×40 magnification using a Zeiss Axio Observer 7 and ZEN (Zeiss)

software. To quantify nuclear levels of p53 and levels of STAT3 in the entire cells and nuclei, Fiji software was used. Briefly, cells were segmented based on thresholding of total signal (STAT3) using the watershed function and nuclear signal using DAPI segmentation. This enables generation of masks, which are used to quantify the signal in the given channel (the entire cell or the nucleus). Images were obtained from randomly chosen areas on the imaged coverslip from biological triplicates. Each quantified condition represents data obtained from at least 100 cells. For quantifications of IGF2BP2 condensations only transfected cells were counted (EGFP-positive cells – co-expression of EGFP and circHIPK3 from same plasmid backbone) and scored visually for frequency of large (>0.5 μm) condensates. The average frequency of condensates was plotted based on 12 (control) and 13 (circHIPK3 OE) different frames, respectively, and error bars represents standard error of mean. p-value was calculated by *T*-test.

## ASO experiment

Antisense oligos were co-transfected into FL3 cells with pcDNA3-Twin-Strep-Tag-IGF2BP2 expression plasmid using Lipofectamine 2000 transfection reagent (Thermo) according to the manufacturer's protocol for combined DNA/siRNA transfection. Briefly, $1.65 \times 10^6$ FL3 cells were seeded in a P10 dish and transfected 24 hr later with 6 μg total DNA (2 μg pcDNA3-Twin-Strep-Tag-RBP expression vector and 6 μg pcDNA3 PL) and 20 nM final concentration of ASO. Cells were incubated for another 48 hr prior to Twin-Strep-tag-RBP RIP as described above.

ASOs used in the study: 11-mer ASO, 5'-mC*mA*T*mC*G*C*  T*G*mG*G*mG*G*mC*C*T* T*mC*T*mA*mG-3' Control ASO, 5'-mC*mG*T*A*mG*G*T*mG*G*C*A*T*mC*mG*C*mC*C*T*mC* mG-3'. m = 2'-*O*-methyl RNA base *Phosphorothioate bond.

## circHIPK3 KD and overexpression

For circHIPK3 KD (time-series excluded, see above), human cancer cell lines FL3, HCT-116, UMUC3, and J82 were grown in DMEM high glucose supplemented with 10% FBS and 1% penicillin/streptomycin (P/S). Cells (HCT-116, J82, UMUC3, and FL3) were seeded in a 12-well plate (1 ml per well), 6-well plate (2 ml per well), or P6 dish (4 ml per well) – approximately 20% confluent at the day of transfection. The volumes and amounts mentioned below cover transfections made in 12-well plates. Double volumes were used for transfection in 6-well plates, 4× volumes in P6 dishes and 10× volumes in P10 dishes. Cells were transfected with siRNA 24 hr later using SilentFect (Bio-Rad): 37.5 μl OptiMEM (Gibco) + 1.25 μl SilentFect (Bio-Rad) was incubated at RT for 5 min before mixing with 1.1 μl siRNA (20 μM stock) pre-diluted in 37.5 μl OptiMEM. After gentle mixing by pipetting, the solution was incubated for 20 min at RT prior to dropwise addition to cells (final concentration of 20 nM). Cells were incubated for 48 hr before harvest. For overexpression studies 2.5 μl Lipofectamine 2000 (Thermo Fisher Scientific) and 0.5–1.0 μg total DNA pcDNA3.1(+)-Laccase2-circHIPK3 or pcDNA3.1(+)-Laccase2-MCS was used. To assess the circ-to-linear relationship in pcDNA3.1(+)-Laccase2-driven circHIKP3 overexpression, two primer sets for the linear version (Lin-HIPK3_1 and Lin-HIPK3_2, *Supplementary file 1*) of the Laccase2-derived precursor were designed and used alongside a divergent primer set for circHIPK3 in RT-qPCR. Standard curves with different amounts of RNA/cDNA were used to verify primer efficiencies.

For RNA purification, cells were harvested by addition of 1 ml Trizol (Thermo Fisher Scientific) followed by mixing. RNA was purified according to the manufacturer's protocol with an additional chloroform extraction to increase quality.

For cDNA synthesis and qPCR, ≥1 μg of total RNA was DNase I (EN0525, Thermo Fisher Scientific) treated according to the manufacturer's description. Between 500 and 800 ng total RNA was used for first strand cDNA synthesis following the description of the Maxima First Strand cDNA Synthesis Kit for RT-qPCR (K1671, Thermo Fisher Scientific). The cDNA concentration for each sample was diluted to 5.0–2.0 ng/μl. Per well of an Agilent skirted plate, 7.5 μl Platinum SYBR Green qPCR SuperMix-UDG (Thermo Fisher Scientific) was mixed with 1 μl cDNA (5.0–2.0 ng/μl), 6.25 μl nuclease-free $H_2O$, and 0.125 μl of each primer (10 μM) to a total reaction volume of 15 μl. Technical triplicates were made for each target gene. Samples were run on an AriaMx Real-Time PCR System and data were analyzed according to the X(0) method described elsewhere (*Thomsen et al., 2010*).

## Quantification of IGF2BP2 copy number

IGF2BP2 copy numbers were measured by comparing signal intensity in western blotting experiments using known amounts of purified IGF2BP2 protein along with cell lysates from a known number of FL3, UMUC3, and J82 cells. Twin-Streptagged IGF2BP2 was expressed in HEK293 Flp-In T-Rex cellsInvitrogen by tetracycline induction (200 ng/ml) for 36 hr prior to cell lysis using hypotonic lysis buffer: 50 mM Tris–HCl pH 7.5, 10 mM NaCl, 5 mM $MgCl_2$, 0.5% Triton X-100, 1% Hexane-1,6-diol, 1 pill Complete – protease inhibitor EDTA-free (Roche) pr. 10 ml. Cells were scraped off and transferred to an Eppendorf tube. The lysate was added NaCl (final concentration of 150 mM), mixed and spun (18,000 × $g$, 15 min at 4°C). The supernatant was incubated with pre-equilibrated MagStrep type 3 XT beads (IBA Life sciences) and rotated for ≥2 hr at 4°C. The magnetic beads were washed 8× with 1.5 ml wash buffer (10 mM Tris–HCl pH 7.5, 150 mM NaCl, 5 mM $MgCl_2$, 0.1% Triton X-100). Twin-Streptagged IGF2BP2 was eluted by boiling beads 90°C, 5 min in 100 µl 2× SDS load buffer. Purified IGF2BP2 concentration was quantified and diluted to 100 ng/µl sample based on comparison with Coomassie staining (GelCode blue (Thermo scientific)) of known amounts of bovine serum albumin.

## Quantification of circHIPK3 copy number

$0.8 \times 10^6$ FL3, J82, or UMUC3 cells were harvested in TRIzol Reagent (Invitrogen, 15596018) and RNA was isolated according to the manufacturer's protocol. DNase treatment and cDNA synthesis were carried out using the Maxima H Minus cDNA Synthesis Master Mix, with dsDNase (Thermo Fisher Scientific, M1682) according to the manufacturer's protocol. qPCR was performed with cDNA corresponding to 0.033% of the total amount of isolated RNA, 2 pmol of each of the primers 5′-GGTCGGCCAGTCATGTATCA-3′ and 5′-ACACAACTGCTTGGCTCTACT-3′, and Platinum SYBR Green qPCR SuperMix-UDG (Thermo Fisher Scientific, 11733038) according to the manufacturer's protocol. circHIPK3 copy number per cell was calculated based on a standard curve made from serial dilutions of a DNA fragment (Twist Bioscience) carrying the sequence amplified by the circHIPK3 primers used for qPCR and starting at $2.5 \times 10^7$ molecules per qPCR reaction.

## BC patient cohorts

Previously analyzed BC cohorts were mapped as described in *Hedegaard et al., 2016*. For new cohorts (*Strandgaard et al., 2022*; *Taber et al., 2020*), RNA was paired-end sequenced (2 × 151 bp) using an Illumina NextSeq 500 instrument. Reads were demultiplexed using bcl2fastq v2.20.0.422 trimmed for traces of adapters using Trim Galore v0.4.1, and mapped to the hg19 genome build using tophat v2.1.1. Gene expression was estimated using cufflinks v2.1.1 and HTseq v0.6.1. CircRNA expression was quantified using the CIRI2 pipeline as described above for HepG2 and K562.

## Statistical analyses

All statistical tests were performed in R (*R Core Team, 2019*; *R Studio Team, 2016*). *DESeq2* was used for differential expression analyses of time-series circHIPK3 KD data (see above) and for RBP KD data. Fisher's exact test was used to evaluate the significance of overlap between two gene sets. The Chi-square test was used to evaluate different enrichments in downregulated, upregulated, and non-DE genes. The non-parametric Wilcoxon Rank Sum Test was used to evaluate differences between groups in large BC patient cohorts. The *T*-test was used to evaluate differences between conditions in wet-lab experiments. For multiple testing correction, we used Benjamini–Hochberg correction and statistical differences were declared significant at FDR <0.1. When multiple testing was not applied, statistical differences were declared significant at $p < 0.05$. Most plots were produced with the R package *ggplot2* (*Wickham, 2016*). The R packages *ggfortify* (*Tang et al., 2016*) and *ggalt* were used to produce PCA plots and *GGally* were used to correlate samples. The R package *VennDiagram* was used to produce Venn diagrams. *Survival* (*Therneau, 2015*; *Therneau and Grambsch, 2000*) and Survminer (*Kassambara et al., 2019*) were used to produce Kaplan–Meier plots and curves were compared statistically by the Log-Rank Test. Additionally, the packages *dplyr* and *reshape* were used to analyze data.

## Acknowledgements

We thank Jeetayu Biswas and Robert Singer for the list of genes with IGF2BP2 KH34 RE (*Biswas et al., 2019*). Flow cytometry was performed at the FACS Core Facility, Aarhus University, Denmark. We thank Micki Grünzig and Karina Hjorth for technical assistance.

## Additional information

### Competing interests

Eugene Yeo: Reviewing editor, *eLife*. The other authors declare that no competing interests exist.

### Funding

| Funder | Grant reference number | Author |
|---|---|---|
| Lundbeck Foundation | R191-2015-1515 | Trine Line Hauge Okholm Morten Muhlig Nielsen Jakob Skou Pedersen |
| Danish Cancer Society | R124-A7869 | Trine Line Hauge Okholm Morten Muhlig Nielsen Jakob Skou Pedersen |
| Danish Council for Independent Research, Medical Sciences | DFF - 7016-00379 | Trine Line Hauge Okholm Morten Muhlig Nielsen Jakob Skou Pedersen |
| Novo Nordisk Fonden | NNF18OC0053222 | Trine Line Hauge Okholm Morten Muhlig Nielsen Jakob Skou Pedersen |
| Harboefonden | 19110 | Trine Line Hauge Okholm Morten Muhlig Nielsen Jakob Skou Pedersen |
| Aage og Johanne Louis-Hansens Fond | 19-2B-503 | Trine Line Hauge Okholm Morten Muhlig Nielsen Jakob Skou Pedersen |
| Carlsbergfondet | CF19-0493 | Trine Line Hauge Okholm |
| Danish Cancer Society | R167-A11105 | Andreas Bjerregaard Kamstrup Anne Kruse Hollensen Mette Laugesen Graversgaard Matilde Helbo Sørensen Christian Kroun Damgaard |
| Lundbeck Foundation | R370-2021-858 | Andreas Bjerregaard Kamstrup Anne Kruse Hollensen Mette Laugesen Graversgaard Matilde Helbo Sørensen Christian Kroun Damgaard |
| Carlsbergfondet | CF20-0236 | Andreas Bjerregaard Kamstrup Anne Kruse Hollensen Mette Laugesen Graversgaard Christian Kroun Damgaard |
| Carlsbergfondet | CF18-0212 | Andreas Bjerregaard Kamstrup Anne Kruse Hollensen Mette Laugesen Graversgaard Christian Kroun Damgaard |
| Dagmar Marshalls Fond | 27126 | Anne Kruse Hollensen Christian Kroun Damgaard |

| Funder | Grant reference number | Author |
| --- | --- | --- |
| National Institutes of Health | R01 HG004659 | Samuel S Park<br>Eugene Yeo |
| National Institutes of Health | U24 HG009889 | Samuel S Park<br>Eugene Yeo |
| Paul G. Allen Family Foundation | Allen Distinguished Investigator Award | Eugene Yeo |
| Paul G. Allen Family Foundation | Paul G. Allen Frontiers Group Advised Grant | Eugene Yeo |

The funders had no role in study design, data collection, and interpretation, or the decision to submit the work for publication.

## Author contributions

Trine Line Hauge Okholm, Conceptualization, Data curation, Software, Formal analysis, Validation, Investigation, Visualization, Writing – original draft, Writing – review and editing; Andreas Bjerregaard Kamstrup, Lasse Sommer Kristensen, Formal analysis, Investigation, Writing – review and editing; Morten Muhlig Nielsen, Software, Formal analysis, Investigation, Methodology, Writing – review and editing; Anne Kruse Hollensen, Data curation, Formal analysis, Validation, Investigation, Visualization, Methodology, Writing – review and editing; Mette Laugesen Graversgaard, Matilde Helbo Sørensen, Formal analysis, Investigation, Visualization; Søren Vang, Formal analysis, Investigation; Samuel S Park, Data curation, Formal analysis, Investigation, Visualization, Methodology; Eugene Yeo, Data curation, Formal analysis, Supervision, Validation, Visualization, Methodology, Writing – review and editing; Lars Dyrskjøt, Data curation, Supervision, Investigation, Writing – review and editing; Jørgen Kjems, Supervision, Writing – review and editing; Jakob Skou Pedersen, Conceptualization, Resources, Data curation, Software, Formal analysis, Supervision, Funding acquisition, Investigation, Methodology, Writing – original draft, Project administration, Writing – review and editing; Christian Kroun Damgaard, Conceptualization, Resources, Data curation, Formal analysis, Supervision, Funding acquisition, Validation, Investigation, Visualization, Methodology, Writing – original draft, Project administration, Writing – review and editing

## Author ORCIDs

Anne Kruse Hollensen (iD) http://orcid.org/0000-0002-5461-6893
Lasse Sommer Kristensen (iD) http://orcid.org/0000-0002-5980-7939
Samuel S Park (iD) http://orcid.org/0000-0002-9230-7661
Lars Dyrskjøt (iD) http://orcid.org/0000-0001-7061-9851
Jakob Skou Pedersen (iD) https://orcid.org/0000-0002-7236-4001
Christian Kroun Damgaard (iD) https://orcid.org/0000-0003-4940-0868

Reviewer #1 (Public Review): https://doi.org/10.7554/eLife.91783.5.sa1
Reviewer #2 (Public Review): https://doi.org/10.7554/eLife.91783.5.sa2
Reviewer #3 (Public Review): https://doi.org/10.7554/eLife.91783.5.sa3
Author response https://doi.org/10.7554/eLife.91783.5.sa4

# Additional files

## Supplementary files

- Supplementary file 1. qPCR primers for detecting circHIPK3.
- Supplementary file 2. Conserved miRNA-binding sites in circHIPK3 based on AgoClip data.
- Supplementary file 3. miRNA expression in J82.
- Supplementary file 4. 7-mer motifs in circHIPK3 and motif scores from RegMex analysis based on gene regulations 24 hr post-transfection in UMUC3 and 48 hr post-transfection in J82.
- Supplementary file 5. Subcellular localization of circHIPK3–RBPs in HepG2.
- Supplementary file 6. DESeq2 output from J82 cells.
- Supplementary file 7. DESeq2 output from UMUC3 cells.

- MDAR checklist

## Data availability

All generated sequencing data have been deposited in NCBI's Gene Expression Omnibus. The circHIPK3 time-series knockdown sequencing data in UMUC3 and J82 cells are accessible through accession number GSE148824. Sensitive personal data (raw data from RNA-Seq and clinical information) cannot be shared publicly because of Danish legislation regarding sharing and processing of sensitive personal data. The publicly available datasets supporting the conclusions of this article, for example, eCLIP data, IGF2BP2 knockdown data, and total RNA-Seq data from HepG2 and K562, etc., are available in the ENCODE repository, https://www.encodeproject.org/, as described in the Methods sections. Output files from DESeq2 analyses are available in *Supplementary files 6 and 7*. All data generated or analyzed during this study are included in the manuscript and supporting files.

The following dataset was generated:

| Author(s) | Year | Dataset title | Dataset URL | Database and Identifier |
|---|---|---|---|---|
| Line T, Okholm H | 2021 | RNA-Seq of circHIPK3 knockdown (KD) samples in UMUC3 and J82 | https://www.ncbi.nlm.nih.gov/geo/query/acc.cgi?acc=GSE148824 | NCBI Gene Expression Omnibus, GSE148824 |

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
