## [Editor Report · eLife assessment]

This work explores the role of one the most abundant circRNAs, circHIPK3, in bladder cancer cells, showing with **convincing** data that circHIPK3 depletion affects thousands of genes and that those downregulated (including STAT3) share an 11-mer motif with circHIPK3, corresponding to a binding site for IGF2BP2. The experiments demonstrate that circHIPK3 can compete with the downregulated mRNAs targets for IGF2BP2 binding and that IGF2BP2 depletion antagonizes the effect of circHIPK3 depletion by upregulating the genes containing the 11-mer. These **important** findings contribute to the growing recognition of the complexity of cancer signaling regulation and highlight the intricate interplay between circRNAs and protein-coding genes in tumorigenesis.

---

## [Referee Report · Reviewer #1 (Public Review)]

In this work the authors propose a new regulatory role for one of the most abundant circRNAs, circHIPK3. They demonstrate that circHIPK3 interacts with an RNA binding protein (IGF2BP2), sequestering it away from its target mRNAs. This interaction is shown to regulate the expression of hundreds of genes that share a specific sequence motif (11-mer motif) in their untranslated regions (3'-UTR), identical to one present in circHIPK3 where IGF2BP2 binds. The study further focuses on the specific case of STAT3 gene, whose mRNA product is found to be downregulated upon circHIPK3 depletion. This suggests that circHIPK3 sequesters IGF2BP2, preventing it from binding to and destabilizing STAT3 mRNA. The study presents evidence supporting this mechanism and discusses its potential role in tumor cell progression. These findings contribute to the growing complexity of understanding cancer regulation and highlight the intricate interplay between circRNAs and protein-coding genes in tumorigenesis.

Strengths:

The authors show mechanistic insight into a proposed novel "sponging" function of circHIPK3 which is not mediated by sequestering miRNAs but rather a specific RNA binding protein (IGF2BP2). They address the stoichiometry of the molecules involved in the interaction, which is a critical aspect that is frequently overlooked in this type of study. They provide both genome-wide analysis and a specific case (STAT3) which is relevant for cancer progression. Overall, the authors have significantly improved their manuscript in their revised version.

Weaknesses:

There are seemingly contradictory effects of circHIPK3 and STAT3 depletion in cancer progression. However, the authors have addressed these issues in their revised manuscript, incorporating potential reasons that might explain such complexity.

---

## [Referee Report · Reviewer #2 (Public Review)]

The manuscript by Okholm and colleagues identified an interesting new instance of ceRNA involving a circular RNA. The data are clearly presented and support the conclusions. Quantification of the copy number of circRNA and quantification of the protein were performed, and this is important to support the ceRNA mechanism.

This is the second rebuttal and the authors further improved the manuscript. The data are of interest to the large spectrum of readers of the journal.

Comments on revision:

The authors explain that they have compared primer efficiencies of two linear Laccase version amplicons and their divergent primers targeting circHIPK3 using amplification standard curves (not shown). They claim that all amplicons were found to be directly comparable, ensuring that their estimation of cirRNA:lineal ratio estimation by RT-qPCR was accurate. I agree that this is not a technically trivial experiment. However, for this measurement to be valid, it is not enough to compare the efficiencies of primers using cDNA/DNA standard curves in the context of the qPCR reaction alone. Instead, one should perform the full RT-qPCR tandem reactions in the context of standard curves of the specific RNAs (for example, obtained by in vitro synthesis). RNA absolute amounts in these standard curves should be known in order to compare the different RNA species (linear or circular).

I do not have major concerns about this issue.

---

## [Referee Report · Reviewer #3 (Public Review)]

Summary:

In Okholm et al., the authors evaluate the functional impact of circHIPK3 in bladder cancer cells. By knocking it down and performing an RNA-seq analysis, the authors found a thousand deregulated genes which look unaffected by miRNAs sponging function and that are, instead, enriched for a 11-mer motif. Further investigations showed that the 11-mer motif is shared with the circHIPK3 and able to bind the IGF2BP2 protein. The authors validated the binding of IGF2BP2 and demonstrated that IGF2BP2 KD antagonizes the effect of circHIPK3 KD and leads to the upregulation of genes containing the 11-mer. Among the genes affected by circHIPK3 KD and IGF2BP2 KD, resulting in downregulation and upregulation respectively, the authors found STAT3 gene which also consistently leads to the concomitant upregulation of one of its targets TP53. The authors propose a mechanism of competition between circHIPK3 and IGF2BP2 triggered by IGF2BP2 nucleation, potentially via phase separation.

Strengths:

The number of circRNAs continues to drastically grow however the field lacks detailed molecular investigations. The presented work critically addresses some of the major pitfalls in the field of circRNAs and there has been a careful analysis of aspects frequently poorly investigated. The time-point KD followed by RNA-seq, investigation of miRNAs-sponge function of circHIPK3, identification of 11-mer motif, identification and validation of IGF2BP2, and the analysis of copy number ratio between circHIPK3 and IGF2BP2 in assessing the potential ceRNA mode of action has been extensively explored and, comprehensively convincing.

Weaknesses:

The authors addressed the majority of the weak points raised initially. However, the role played by the circHIPK3 in cancer remains elusive and not elucidated in full in this study.

Overall, the presented study surely adds some further knowledge in describing circHIPK3 function, its capability to regulate some downstream genes, and its interaction and competition for IGF2BP2. However, whereas the experimental part sounds technically logical, it remains unclear the overall goal of this study and the achieved final conclusions.

This study is a promising step forward in the comprehension of the functional role of circHIPK3. These data could possibly help to better understand the circHIPK3 role in cancer.

---

## [Author Response]

The following is the authors’ response to the previous reviews.

**Reviewer #1 (Recommendations For The Authors):**
Major points about revised manuscript(1) While I acknowledge that the Laccase2 vector is probably the best available in terms of its clean circRNA-expression potential, the authors still lack an estimation of the circRNA overexpression efficiency, specifically the circular-to-linear expression ratio. In their second rebuttal letter, the authors argue that they do not have the option to use another probe and that they are limited by the Backsplicing junction (BSJ)-specific one. I assume they mean that such a probe might only partially hybridize with the linear form and therefore give a poor or no signal in the Northern blot. However, in this referee's opinion, it is precisely because of this limitation that the authors should have used another probe against both the linear and circular RNAs to simultaneously and quantitatively detect both isoforms. This would have allowed them to reliably estimate a circular-to-linear ratio. Perhaps the linear isoform is indeed not expressed or is very low for this circRNA overexpression vector, but the probe used by the authors does not prove it. I think that this addition to the manuscript is not strictly necessary at this stage, but it would certainly improve the results.

We fully agree with this recommendation. Our efforts to show this using northern blotting was unfortunately unsuccesful due to background signal. To accommodate the question about circ-to-linear ratio, we instead used an RT-qPCR strategy to measure the linear vs circRNA expression derived from the LaccasecircHIPK3 expression constructs/cell lines. To be able to compare obtained results from different amplicons, we measured primer efficiencies (using amplification standard curves – not shown) of two linear Laccase version amplicons and our divergent primers targeting circHIPK3, which were found to be directly comparable. Using these primer sets in RT-qPCR on the same RNA preparation (total cellular RNA) from the northern blot (Supplementary figure S5H) revealed a ~4 fold higher expression of circHIPK3 compared to linear precursor RNA (Supplementary Figure S5I).

This demonstrates that the Laccase vector system efficiently produces circHIPK3 RNA as expected.

The few changes to the manuscript (results section text and reference to Supplementary Figure S5I) has been highlighted in yellow. The materials and methods section and Table S1 have been modified to include description of RTqPCR and specific primers.